



# In-flight calibration results of the TROPOMI payload on-board the Sentinel-5 Precursor satellite

Antje Ludewig[1], Quintus Kleipool[1], Rolf Bartstra[1,3], Robin Landzaat[1,2], Jonatan Leloux[1,2], Erwin Loots[1], Peter Meijering[1,2], Emiel van der Plas[1], Nico Rozemeijer[1,2], Frank Vonk[1,2], and Pepijn Veefkind[1]

[1]KNMI, Royal Netherlands Meteorological Institute, De Bilt, The Netherlands
[2]TriOpSys B.V., Utrecht, The Netherlands
[3]S&T Science and Technology B.V., Delft, The Netherlands

**Correspondence:** ludewig@knmi.nl

**Abstract.** After the launch of the Sentinel-5 Precursor satellite on 13 October 2017 its single payload, the Tropospheric Monitoring Instrument (TROPOMI), was commissioned during 6 months. In this time the instrument was tested and calibrated extensively. During this phase the geolocation calibration was validated using a dedicated measurement zoom mode. With the help of spacecraft manoeuvres the solar angle dependence of the irradiance radiometry was calibrated for both internal

diffusers. This improved the results that were obtained on-ground significantly. Furthermore the orbital and long term stability was tested for electronic gains, offsets, non-linearity, the dark current and the output of the internal light sources. The CCD output gain of the UV, UVIS and NIR detectors shows drifts over time which can be corrected for in the L1b processor. In-flight measurements also revealed inconsistencies of the radiometric calibration and degradation of the UV spectrometer. Degradation is also detected for the internal solar diffusers. Since the start of the nominal operations (E2) phase in orbit 2818

on 30 April 2018, regularly scheduled calibration measurements on the eclipse side of the orbit are used for monitoring and updates to calibration key data. This article reports on the main results of the commissioning phase, the in-flight calibration and on the instrument's stability since launch. Insights from commissioning and in-flight monitoring led to updates to the Level 1b processor and its calibration key data. The updated processor is planned to be used for nominal processing from 2020 on.

## 1 Introduction

The Sentinel-5 Precursor (S5P) mission is part of the Copernicus Earth observation programme by the European Union. It is the first atmospheric observing mission within this programme (Ingmann et al., 2012). With its launch on 13 October 2017 the S5P mission can avoid large gaps in the availability of global atmospheric products between the future missions Sentinel-4 and Sentinel-5 and earlier and on-going missions such as SCIAMACHY (Bovensmann et al., 1999), GOME-2 (Munro et al., 2016), and OMI (Levelt et al., 2006). The S5P satellite flies in a low Earth orbit (824 km) and is Sun-synchronous with an

equator crossing time of 13:30 local solar time. The TROPOspheric Monitoring Instrument (TROPOMI) is the only payload on S5P. It was jointly developed by the Netherlands and ESA. With its push-broom imaging system with spatial sampling down to 5.6 km×3.6 km a daily global coverage is achieved for trace gases and aerosols important for air quality, climate forcing, and the ozone layer. TROPOMI contains four spectrometer with spectral bands in the ultraviolet (UV), the visible (UVIS),



**Table 1.** Main products and characteristics of the four TROPOMI spectrometers and the definition of the spectral bands with identifiers 1–8. The listed values are based on on-ground calibration measurements (see Kleipool et al. (2018)) and are valid at the detector centre. The performance range is the range over which the requirements are validated, the full range is larger. The nominal spatial sampling distance (SSD) is given at nadir for the updated operations scenario.

| Spectrometer | UV | | UVIS | | NIR | | SWIR | |
|---|---|---|---|---|---|---|---|---|
| Band ID | 1 | 2 | 3 | 4 | 5 | 6 | 7 | 8 |
| Main level 2 products | $O_3$ | | $O_3$, $SO_2$,$CH_2O$, aerosols $NO_2$, clouds | | aerosols, clouds | | CO, $CH_4$ | |
| Full spectral range [nm] | 267–300 | 300–332 | 305–400 | 400–499 | 661–725 | 725–786 | 2300–2343 | 2343–2389 |
| Performance range [nm] | 270–320 | | 320–490 | | 710–775 | | 2305–2385 | |
| Spectral resolution [nm] | 0.45–0.5 | | 0.45–0.65 | | 0.34–0.35 | | 0.227 | 0.225 |
| Spectral sampling [nm] | 0.065 | | 0.195 | | 0.125 | | 0.094 | |
| Nominal SSD [$km^2$] | $28.8 \times 5.6$ | $3.6 \times 5.6$ | $3.6 \times 5.6$ | $3.6 \times 5.6$ | $3.6 \times 5.6$ | $3.6 \times 5.6$ | $7.2 \times 5.6$ | $7.2 \times 5.6$ |
| Row binning factor | 16 | 2 | 2 | 2 | 2 | 2 | n/a | n/a |

the near-infra-red (NIR), and the shortwave infra-red (SWIR) wavelengths (Veefkind et al., 2012). The main characteristics of

TROPOMI are listed in Table 1.

This wavelength range allows for observation of key atmospheric constituents such as ozone ($O_3$), nitrogen dioxide ($NO_2$), carbon monoxide (CO), sulfur dioxide ($SO_2$), methane ($CH_4$), formaldehyde ($CH_2O$) aerosols and clouds. The instrument is measuring the radiance on the day side of each orbit and once a day the irradiance via a dedicated solar port as described in detail in KNMI (2017) and Kleipool et al. (2018).

The S5P mission is flying in constellation with the NOAA/NASA Suomi NPP (National Polar-orbiting Partnership) satellite. The difference in overpass time is 3–5 min, so high resolution cloud information and vertically resolved stratospheric ozone profiles from its instruments VIIRS (Visible Infra-red Imaging Radiometer Suite) and OMPS (Ozone Mapping and Profiler Suite) can act as supplementary input for TROPOMI data processing. Prior to launch the instrument was tested and calibrated as reported in Kleipool et al. (2018). Not all calibration data could be derived with the desired accuracy and had to be recovered

during the E1 phase. For the solar angle dependence of the irradiance radiometry this could be done during the commissioning phase.

During the first 6 months of the mission the instrument was commissioned and dedicated measurements were scheduled to validate the geolocation, calibrate the angular dependency of the irradiance radiometry for both internal diffusers and calibrate detector and electronic effects such as gain, offsets and non-linearity. All instrument settings for all internal and external

sources were checked and optimised for optimal signal to noise while leaving margin for changes in signal. The timing and definition of the different orbit types was adapted to match the detected darkness of the eclipse. The changes to the instrument settings and on-board procedures were extensively tested and burnt into the instrument's electrically erasable programmable read-only memory (EEPROM) before the start of the nominal operations (E2) phase. In regular intervals dedicated monitoring





measurements were performed to assess the instrument's long term stability and its stability over an orbit. Also radiance
and irradiance data was measured to optimise the nominal settings and allow for a testing of the S5P Payload Data Ground
Segment (PDGS) and L2 processing. In two different zoom modes also high spatial sampling radiance data was measured for
$NO_2$, clouds, $CH_4$ and CO retrievals.

Since orbit 2818 on 30 April 2018 the mission is in its nominal operations (E2) phase with a fully repetitive scenario and
systematic processing and archiving of data products by the PDGS. The L2 products are disseminated to both operational users
(e.g. Copernicus services, national Numerical Weather Prediction (NWP) centres, value adding industry) and the scientific user
community. The repetitive scenario includes daily solar measurements and calibration measurements with internal light sources
at the eclipse side of each orbit. In the following, the main results from the commissioning phase and in-flight monitoring will
be presented.

## 2   Thermal stability

At the very beginning of the mission the instrument prime, Airbus Defence and Space Netherlands, could confirm that the
thermal controls are within their predicted values and that the temperature setpoints can remain the same as used during
on-ground calibration, see Kleipool et al. (2018). According to the prediction there is sufficient residual margin on all active
thermal control channels to ensure temperature stability of the complete instrument over the entire mission lifetime. Monitoring
shows that the detector temperatures are stable within 10–30 mK, the lower values are for the SWIR and UV detectors. The
NIR and UVIS detectors are within the larger range. The UVN optical bench module (OBM) with the common telescope and
UVN optics as well as the SWIR specific OBM are stable within 60 mK. All values are well within their specifications. During
nominal instrument operation, the only exceptions to the thermal stability occur during and after orbital manoeuvres. The UVN
detectors and OBM recover their stability within 1–2 orbits. For the SWIR grating, the recovery takes the longest time: for
out-of-plane manoeuvres up to 35 orbits have been observed. This leads to an estimated spectral shift of $0.12\pm0.01\,\mathrm{nmK^{-1}}$.
With version 2 of the L1b processor, measurements taken under non-nominal thermal conditions will be flagged.

## 3   Light tightness

The folding mirror mechanism (FMM) closes the Earth port of the instrument and relays light from the calibration unit (CU)
to the instrument's telescope. When the FMM is closed the entire instrument can be closed off from external light for certain
positions of the diffuser mechanism (DIFM). The closed position is however not entirely light tight as in-flight tests showed.
For the UVN detectors signals up to 100 times the dark current could be observed when the instrument is in closed position.
For the SWIR module no light leaks were detected, however the SWIR module is sensitive to hot spots such as gas flares on
the eclipse side (see van Kempen et al. (2019)). The nominal operations baseline was therefore adapted such that all calibration
measurements only start once the spacecraft is in full eclipse and the radiance background is only measured with a closed
FMM, as described in Section 14.





**Table 2.** The observed average approximate decrease in output for the internal light sources per 1000 orbits.

| Source | UV | UVIS | NIR | SWIR |
|--------|------|------|------|------|
| DLED | 0.64 % | 0.59 % | 0.74 % | 0.15 % |
| WLS | 0.90 % | 0.77 % | 0.22 % | 0 % |
| CLED | n/a | 0.33 % | n/a | n/a |

## 4 Internal sources

The TROPOMI instrument contains several internal light sources. LED strings are placed close to each of the detectors (DLED) and in the calibration unit are a white light source (WLS), a LED in the visible wavelength range (CLED) and a spectral line source (SLS). The SLS consists of 5 temperature tunable narrowband diode lasers in the SWIR wavelength range. During the commissioning phase, all internal sources were checked and compared to measurements performed during on-ground calibration. The differences in detector response in-flight relative to on-ground are close to 1 for DLED, CLED, and SLS. The WLS shows the expected increase in brightness due to the micro-gravity environment, the signal is about 1.1–1.4 times larger in-flight. During nominal operations the internal light sources are used for calibration measurements and their output is monitored for ageing effects. For the DLED and the CLED the average detector response decreases approximately linearly. For the WLS the response decreases linearly for all detectors but the SWIR. Both the CLED and WLS show variations in output of ±0.5 % from measurement to measurement, for the DLED the variation is smaller than 0.05 %. The average decrease in output per 1000 orbits is shown in Table 2. The output of the SLS is stable as already reported in van Kempen et al. (2019).

## 5 Orbital electronic stability

The orbital stability of electronic gains, offsets and noise was tested with dedicated measurements during the commissioning phase. For SWIR, as reported in van Kempen et al. (2019), no orbital dependencies were detected for offset, dark current and noise.

For UVN the orbital dependency of the dark current could not be established since the FMM is not sufficiently light tight. The dark current measurements on the eclipse side suggest a dark current of $2\,e^-s^{-1}$, this is consistent with the on-ground results. Also the offsets that are derived from in-flight data show the same behaviour as on-ground. There is no significant orbital dependency of the computed gains for the programmable gain amplifier (PGA), correlated double sampling gain (CDS) and CCD output node gain ratios, however there is a temporal drift of the CCD gain ratio and the gain alignment between bands, see Section 6 below. The non-linearity calibration key data (CKD) obtained from in-flight data differs from the on-ground key data no more than 1 ‰ of the signal.



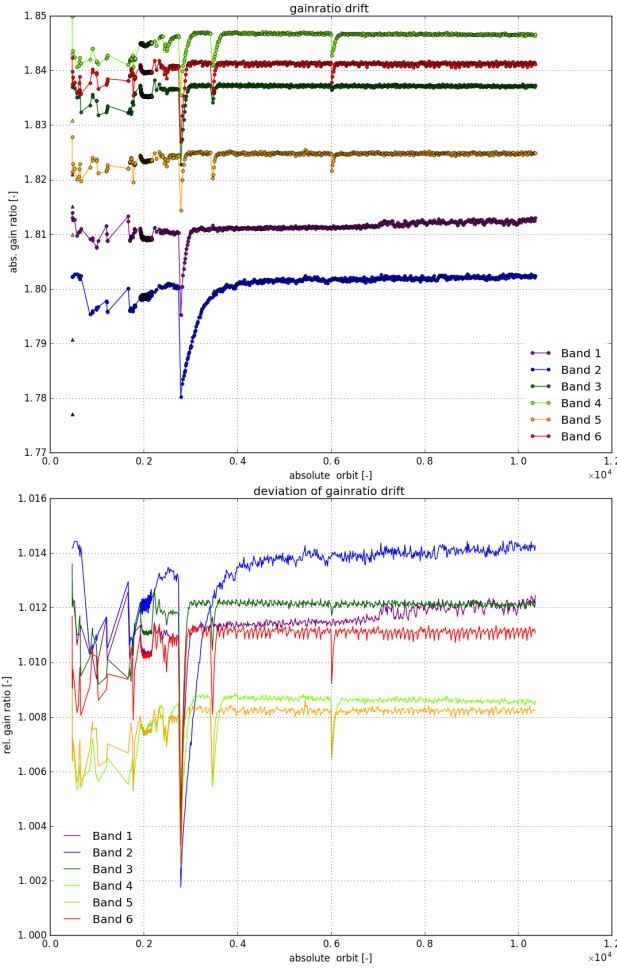

**Figure 1.** Gain drift of the UVN CCD detector output nodes. The top panel shows the ratio of high and low CCD gain setting over time and the lower panel shows this ratio over time with respect to the on-ground ratio. The different colours represent the different detector bands (see legend).



## 6   Gain drifts UVN detectors

The CCD output nodes of the UVN detectors convert the signal from charge to voltage. The CCD output nodes can be used

with a high or a low electronic gain to optimize the signal. It has been found that the amplification can drift in time for the low CCD gain setting. The drift can be calculated from the relative change in the gain ratio between the high and low CCD gain setting. The CCD gain ratio is derived from the image-averaged signals of unbinned DLED measurements with four different exposure times for both high and low CCD gain. Regression lines are fitted through these four data points for each gain setting. The ratio of the slopes of the regression lines for both CCD gain settings is the CCD gain ratio. The ratios are around 1.8, but

are different for each band as shown in the top panel of Fig. 1. The figure shows the variation over time of the ratio of CCD gain settings and in the lower panel of Fig. 1 the ratio with respect to the gain ratio measured on-ground is shown. Compared to the on-ground value the ratio deviation is always at least 0.54 % and reaches more than 1.4 % in band 2. The biggest change occurs around orbit 2765, when the instrument was switched off entirely to be able to burn the EEPROM. Before the EEPROM burn there are large variations and less measurement points. This is during the commissioning phase. Since the EEPROM burn

the measurements are all taken according to the nominal operations baseline, so radiance measurements on the day side and calibration measurements on the eclipse side of each orbit. After the EEPROM burn the gain ratio relaxes at different rates to more constant values, however during spacecraft manoeuvres around orbits 3470 and 6010 the bands 3–6 – the UVIS and NIR detectors – show dips in the gain ratio. During the E2 phase eight other spacecraft manoeuvres were performed which do not show in the gain ratio. No correlations were found between the drift in the gain ratio and changes in temperatures or voltages

of the instrument.

Due to the separate read-out chains for each detector half, the band signals need to be aligned in the centre of the detector as described in KNMI (2017), a drift in the CCD gain also changes this gain alignment. When the gain alignment factor for each band is calculated, the ratio of these factors follows the ratio of the low-high gain ratio drift as can be seen in Fig. 2. The right panel of Fig. 2 shows the drift relative to the first available in-flight measurement. The correlation between the interband

gain ratio drift and the alignment gap becomes clear. The gain alignment for the high CCD gain setting changes by less than 0.05 %, while for the low gain setting changes up to 0.5 % occur. The UVIS and NIR detectors show similar behaviour. During the nominal operations phase E2 the CCD gain ratio is computed on a daily basis from dedicated DLED measurements. The computation is automatically done by the in-flight calibration (ICAL) processor at the payload data ground segment (PDGS) and the result is therefore available in the calibration data product. The correction of the gain drift is done in the L1b processor

with a regularly updated calibration key data (CKD) file. The signal for a specific UVN band is then corrected with the interpolated or extrapolated factor depending on the current orbit number.

After the gain drift correction, the recomputed alignment factor is indeed more or less one, for both the high and the low gain measurement as shown in Fig. 3 and does not follow the initially derived gain ratio drift (blue line). The inter-band alignment gap now stays well below 0.1 % for all orbits and all bands. The deviation from the alignment will be continually monitored.

If the deviations grow in spite of the gain drift correction, a re-alignment can be performed by a CKD update.

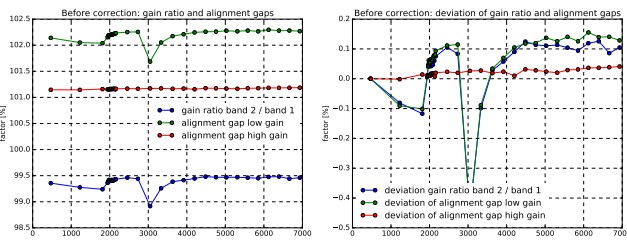

**Figure 2.** Without CCD gain drift correction: the alignment correction factor ratio between bands 1 and 2 for low (green) and high (red) CDD gain setting together with the relative drift of the computed gain ratios (blue). The left panel shows the absolute values and the right panel shows the values relative to the first available in-flight data. The alignment correction factor ratio for low gain follows the gain ratio drifts.

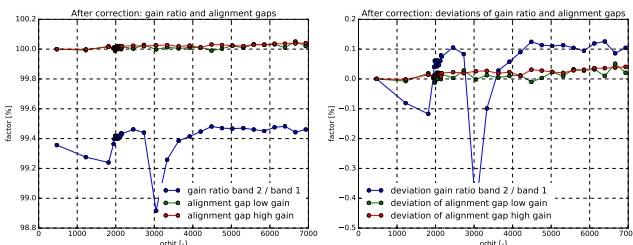

**Figure 3.** With CCD gain drift correction: the alignment correction factor ratio between bands 1 and 2 for low (green) and high (red) CDD gain setting together with the relative drift of the originally computed gain ratios (blue). The left panel shows the absolute values and the right panel shows the values relative to the first available in-flight data. The alignment correction factor ratio is around 1 after the gain drift correction.

The simplest solution to increase the gain stability would be to only use the high CCD gain setting. This is however not possible for radiance measurements. A high CCD gain setting would require shorter exposure times to avoid saturation of amplifiers in the electronic read-out chain. The high optical throughput of the UVIS and NIR spectrometers already require the shortest possible exposure times. For the UV spectrometer the high fixed gain in the analogue video chain and ozone hole

conditions prevent that the high CCD gain can be used. To minimize the possible impact on the values of the Earth's reflectance, it was chosen to use the same instrument settings for both radiance and irradiance measurements where possible.

## 7 Pixel saturation and charge blooming

For very bright radiance scenes, for example above high clouds, the CCD pixels of bands 4 and 6 can saturate. This is caused by the combination of the optical throughput, which is higher than designed, and the pixel and register full well values, which

are lower than designed. By adapting the binning schemes for the CCD detectors and minimizing the exposure time, this could be partly mitigated. However, it is impossible to completely avoid saturation for bands 4 and 6. In case of heavy pixel saturation, charge blooming can occur: excess charge then flows from saturated pixels into neighbouring pixels in the detector





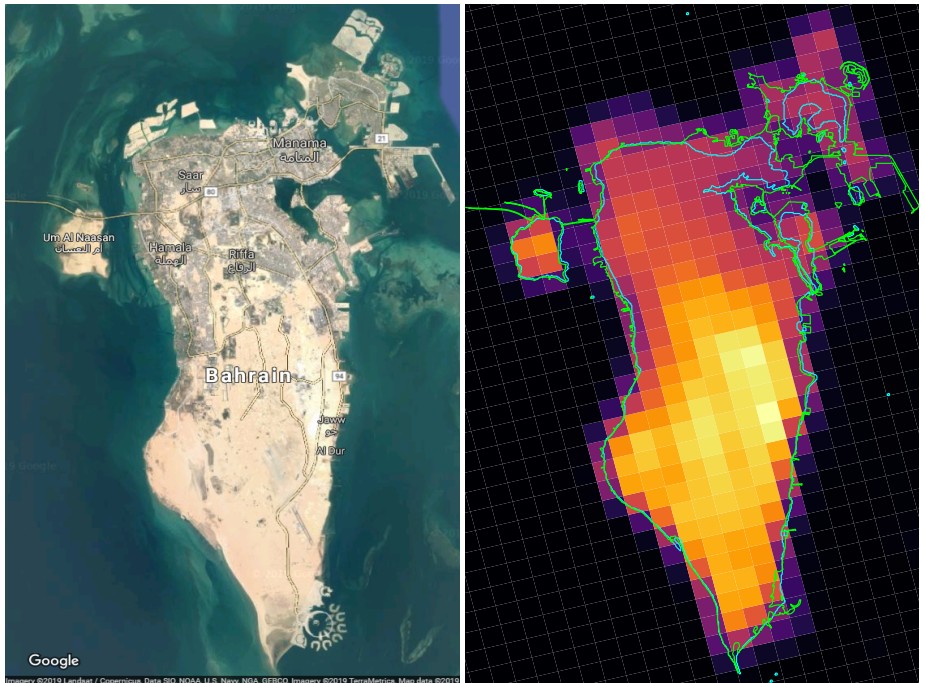

**Figure 4.** The left plot shows a ©Google maps satellite view of the island country of Bahrain. The smaller island Um Al Naasan to the west has a size of approximately $4 \times 5.5$ km. The colourmesh plot on the right hand side is made using TROPOMI geolocation zoom radiance data of band 6 for orbit 1305. The scene is situated at nadir and has a ground pixel size of approximately $1.8 \times 1.8$ km. The two reference coastline datasets described in the text are plotted in light-blue (500 m accurate WVS) and light-green (50 m accurate PGSD). The contrast between water and the desert type land is large. In the north and south-east newly created artificial islands can be seen which are measured by TROPOMI and are visible on ©Google maps but are not included in the coastline references, as these were produced using older satellite measurements.

row direction. For TROPOMI this means that a bright, saturated scene will affect neighbouring scenes, resulting in higher signals for one or more spectral pixel in these neighbouring scenes. The internal light sources are not suitable to observe this

effect, as they have a flat illumination pattern and charge blooming is best observed with a high contrast in the detector row (across-track) direction. Therefore reflectance data from saturated scenes was used to determine the extent of the blooming for various pixel fillings. A new dedicated L1b algorithm checks if pixel fillings exceed specific thresholds and then flags up to 24 pixels in row direction. This new algorithm is included in version 2 of the L1b processor.

## 8 Geolocation

During on-ground calibration the line-of-sight of each TROPOMI detector pixel was calibrated with a collimated white light source as described in Kleipool et al. (2018). As there is no comparable source available in-flight, a special measurement mode was developed for in-flight validation where data is acquired for all detectors with the highest possible spatial resolution. This

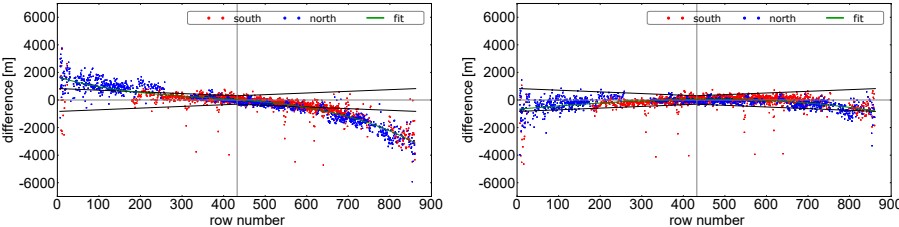

**Figure 5.** Left plot: Along-track distance between the coastline points determined from TROPOMI (band 6) radiance and the 50 m accurate PGSD reference versus illuminated detector row number. The land-water orientation of the scenes are labelled by colour, while the black lines indicate the geolocation knowledge requirements, as extended linearly from nadir to edge-of-swath. Ignoring outliers, at nadir (indicated by a vertical line) the differences lie within the requirement. However, large positive differences for low row numbers and large negative differences for high row numbers are distributed linearly. Right plot: The same data but now with a yaw-angle correction of 0.002 radians applied in the L1b processor. The differences for low and high row numbers are now mostly within the requirements (black lines) and more symmetrical.

is done by setting the binning factor for UVN to 1 for all illuminated rows and reducing the co-addition time for all detectors. This zoom-mode leads for the UVN spectrometers to ground pixels with a size of approximately $1.8 \times 1.8$ km in along-track $\times$

across-track direction at nadir and $1.8 \times 9.2$ km at the edge of the swath. For the SWIR spectrometer $1.8 \times 7.1$ km is reached at nadir and $1.8 \times 37.5$ km at the edge of the swath. However, not all detector pixels can be read out with this high resolution, as both internal data rate limits and the data downlink limit would be reached. To circumvent that, only a small range of columns at the detector edges is read out for the UVN detectors. The SWIR module has a CMOS detector and pixel selection can only be done per band, so it was chosen to read out only band 7, the lower wavelength half of the SWIR detector.

For the analysis a number of latitude-longitude windows are selected with a straight coastline with a large radiance contrast in either across-track or along-rack direction. Within these windows the four consecutive ground pixels with the largest radiance difference are found in the direction orthogonal to the coastline. A third-degree polynomial is fitted through these four points and its inflection point is calculated. If the inflection point lies between the second and third pixel, it is considered to be a measured coastline point by TROPOMI. Scenes where cloud coverage disturbs the coastline determination are discarded by

visual inspection. Two reference coastline datasets published by NOAA are used to determine the difference with the polyline formed by the valid inflection points: the (preliminary) circa 50 metre accurate high-water line prototype global shoreline data (PGSD) (NOAA, 2016a), and the circa 500 metre accurate average-water line world vector shoreline (WVS) (NOAA, 2016b) datasets released within the global self-consistent hierarchical high-resolution geography database (GSHHG). As can be seen on the right in Fig. 4 the differences between the high- and the average-water reference are quite large. Furthermore

it can be seen that the used references are based on out-dated satellite imagery: the artificial island group Durrat Al Bahrain (construction start 2004) in the south-east is only partially visible in the PGSD reference. The accuracy of the available coastline data in combination with the deviating tidal level during the TROPOMI overpass is a source for errors in this analysis. Other possible error sources are shallow water with increased radiance levels, river estuaries, lagoons and clouds missed by the visual





**Table 3.** The wavelength shift $\Delta\lambda$ as implemented in the update of the nominal wavelength annotation CKD. Note that only one value per detector has been chosen.

| Detector | UVIS | | NIR | | SWIR | |
|---|---|---|---|---|---|---|
| Band | 3 | 4 | 5 | 6 | 7 | 8 |
| Observed shift [pm] | 29 | 25 | – | −2.5 | −40 | – |
| Observed shift [pixel] | $\frac{1}{7}$ | $\frac{1}{8}$ | – | $-\frac{1}{50}$ | $-\frac{1}{2}$ | – |
| Implemented $\Delta\lambda$ [pm] | 27 | | 0 | | −40 | |

inspection. The analysis was performed for different scenes distributed all over Earth for bands 4–7, for bands 1–3 the contrast
was found to be too small. The best land–sea contrast is observed for band 6.

The shortest distance between each determined coastline inflection point and a reference coastline polyline is determined, in
longitude and latitude as well as absolute. Using an approximate spacecraft average heading angle of 12° around the equator,
these differences are converted to along-track and across-track distances. The mission requirement on the ground pixel position
knowledge is 305 m at nadir and 825 m (1500 m) at the edge of swath in along-track (across-track) direction. The distance in
along-track direction is shown for band 6 in Fig. 5, the location of the landmass with respect to the sea is indicated in colours.
In the left plot it is clear that the low row numbers, corresponding to the western part of the swath, display a bias towards the
north (positive distance), while the eastern part of the swath (high row numbers) has a bias to the south. This corresponds to
an error in the yaw-angle of the geolocation. For the SWIR and UVIS detectors the same effect is observed, so a mechanical
change within the instrument during launch seems unlikely. The gravity release of the topfloor of the platform could cause
the change in pointing. From the measurements the yaw-angle correction has been been determined to be 0.002 radians. This
correction has been implemented in the L1b processor since version 1 which has been operational before the start of the E2
phase. As can be seen in the right plot of Fig. 5, with the updated geolocation, the along-track differences are symmetrical and
the ground pixel knowledge is mostly within the mission requirements.

## 9 Spectral annotation

The L1b processor assigns a wavelength to every spectral pixel based on on-ground calibration data. In L2 processing this
assignment is used as an initial value for wavelength fitting. After launch it was observed from L2 retrievals that the assignment
is shifted with respect to the fitted values. From gravity release and the connected mechanical relaxation some impact on the
spectral calibration can be expected. For SWIR spectrometer, the temperature of the grating plays a big role for the wavelength
stability. The wavelength fit results from the algorithms for daily aerosol index (band 3), $NO_2$ (band 4), FRESCO (band 6) and
CO (band 7) were used as input for a CKD update. For other bands no operational data, where only a wavelength shift and
no wavelength squeeze is fitted, was available. The wavelength fits showed some variation both over the detectors and over
time. For the UVN spectrometers both variations are within the accuracy of the on-ground calibration values of 9 pm. For the





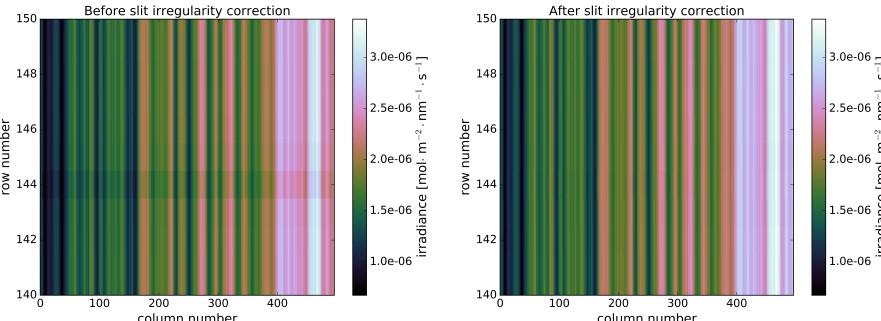

**Figure 6.** Zoom to the rows around the slit irregularity for binned irradiance data with ICID 202 via diffuser QVD1 for band 2 before (left) and after L1b correction (right). Note that the binned row count is shown in the plots, the affected detector rows are rows 335–337. The correction is effective.

SWIR spectrometer, the change over time is much larger than the on-ground accuracy (0.06 pm) and is related to the very long thermalization time of the grating.

The nominal wavelength annotation CKD has been updated with a wavelength shift $\Delta\lambda$ based on radiance L2 fitting data. In Table 3 the averaged observed shifts and the implemented correction to the CKD are shown. A single shift per detector is added to the on-ground calibration data, but only where the shift exceeds the on-ground calibration accuracy. The value has been chosen from data at the middle of September 2018 for SWIR and at the beginning of October 2018 for UVN. The correction will become active with version 2 of the L1b processor. In case the wavelength calibration changes further, the CKD can be updated.

## 10 Slit irregularity

When the slit in the optical path is locally obstructed, the instrument throughput is lowered for specific viewing angles corresponding to detector rows and the instrument spectral response function (ISRF) can change for these angles. For the UV detector a lower signal was observed for detector rows 335–337 after launch. The other detectors show no signature of a slit-irregularity. Therefore not the main instrument slit but the slit in the UV spectrometer is most likely causing the feature. A slit-irregularity correction had already been foreseen in the L1b processor, so only an update of the calibration key data (CKD) was needed. The CKD has been derived from unbinned measurements with the internal white light source (WLS). The WLS is located inside the calibration unit and its light reaches the main instrument telescope via the side of either one of the diffusers and the folding mirror mechanism (FMM). Unlike radiance or irradiance measurements, the WLS provides a smooth spectrum without spectral lines. The image is corrected for the pixel response non-uniformity (PRNU), normalized with the signal in an unaffected row in the vicinity of the irregularity and then fitted linearly over 25 rows around the irregularity. To improve the fit, the signal is averaged over 5 spectral pixels. The derived correction is the largest in detector row 335 with 6 % and has been determined with a relative error of 1.09 % for band 1 and 0.30 % for band 2. The error is larger in band 1 due to a





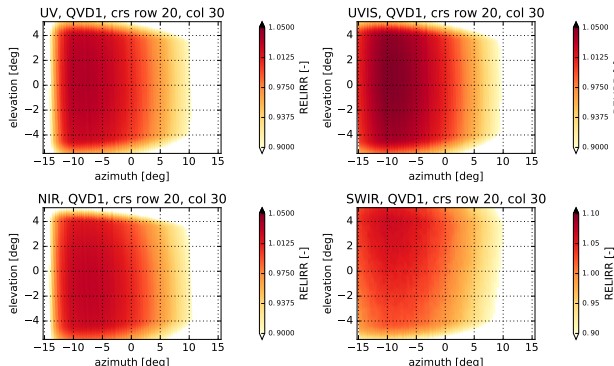

**Figure 7.** The relative irradiance in-flight measurements for diffuser 1 (QVD1) divided by their reference measurements. Shown are the values for the different solar angles for a super pixel in the detector corner (row 20, column 30) for each detector. The variation in azimuth direction is smooth. For both solar angles the signal cut-off is visible.

lower signal-to-noise ratio of the available measurements. Figure 6 shows the irradiance signal in band 2 before and after the

correction. The shown measurement type with instrument configuration identifier (ICID) 202 is the one also used for Level 2 processing with the same binning scheme as the nominal radiance measurements. Detector rows 335 and 336 correspond in this example to the binned row counter 144. The slit-irregularity is so far stable, both in location and magnitude. The stability and the remaining effects were determined using corrected WLS measurements from different instrument settings and from different orbits during mission. The validation confirms the uncertainty as derived for the CKD error. The on-board light sources

for the UVN spectrometers are not suitable to investigate a possible change of the ISRF for the affected rows. However, most Level 2 algorithms take small changes of the ISRF into account, so the impact is expected to be small. The slit-irregularity correction will become active with version 2 of the L1b processor.

## 11   Relative irradiance calibration

The relative angular radiometry of the TROPOMI solar port had been measured during the on-ground calibration campaign.

However the measurement suffered from instabilities of the optical stimulus and as a consequence only key data for one of the two internal quasi volume diffusers (QVD), namely QVD2, could be derived with a reduced angular resolution, see Kleipool et al. (2018). In-flight the entire elevation angle range of the solar port is covered during each solar measurement, however the azimuth angle range is only covered over the course of one year. To obtain valid key data for the entire solar angle range before the start of nominal operations, the different azimuth angles were obtained by moving the platform with a slew manoeuvre in

successive orbits. Both internal solar diffusers QVD1 and QVD2 were re-calibrated with a higher sampling of the illumination angles than used on-ground. For QVD1, the main diffuser, 400 orbits were used for the solar calibration, this corresponds to azimuth angles every 0.15° between -15° and +15° with reference points in between. During on-ground calibration it was not possible to cover the entire azimuth range and the measurement grid was 10 times coarser than in-flight. For the elevation





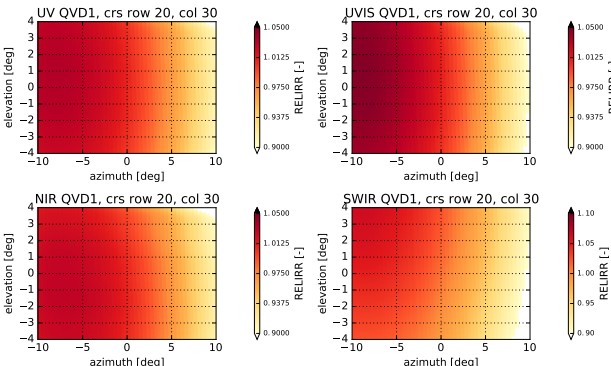

**Figure 8.** The fit for the values in Fig. 7 using an 8th order Chebyshev polynomial both in the azimuth and elevation direction. The polynomial was fitted to values between -10° and +10° in the azimuth direction and -4° and +4° in the elevation direction.

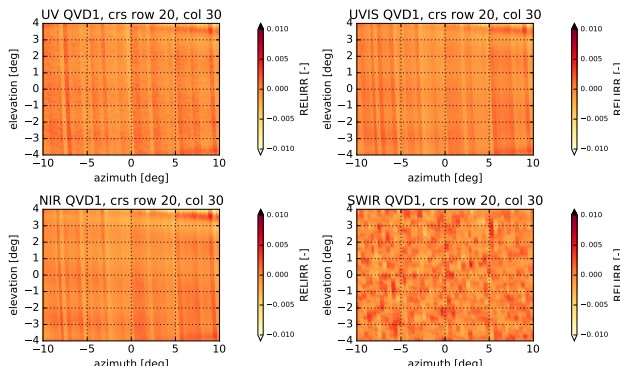

**Figure 9.** The residuals of the fit shown in Fig. 8. Orbit to orbit variations (visible as stripes) form the main contribution.

angle the in-flight grid is more than 25 times finer than on-ground. For QVD2, the backup diffuser, the sampling was reduced

to 0.25° over 240 orbits in the same azimuth range and also with references in between. The reduction was chosen due to the observed degradation in QVD1 (see also Section 12). The reference points are measured to account for instrument degradation and changes in solar output. The reference angle is 1.269° azimuth and 0° elevation, the same solar angle as used on-ground for the absolute irradiance calibration. The solar measurements are performed around the northern day-night terminator, where the solar zenith angle is approximately 90°. During the solar measurements the azimuth angle drifts over a small range ($\approx$1.5°)

around the commanded azimuth angle. The measurement duration is long enough to cover the full elevation range ($\approx$-5°– +5°). From each series of azimuth angles around the reference azimuth angle, the frame closest to the reference angle is chosen as the reference measurement. This frame is then used to determine the relative irradiance and degradation. The overall azimuth grid is sampled such that the full range is scanned several times with a successively finer resolution alternating with reference measurements. This is done to ensure the sampling of the entire solar angle range even if not all measurements can





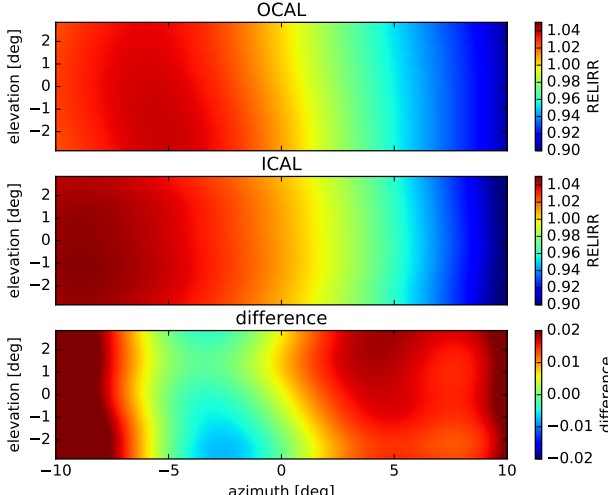

**Figure 10.** The CKD for band 3 for diffuser QVD1 as derived from the on-ground campaign data (top), the CKD derived from commissioning phase data (middle) and the difference between the two (lower panel). Shown is the value for a super pixel in the corner of the detector in row 20 and column 30. The in-flight CKD shows more detail and the CKDs differ up to 2 percent points.

be performed or are missing due to downlink issues. Both QVDs were measured without row binning in the illuminated region. The CKDs for QVD1 and QVD2 do not differ substantially, therefore only results for QVD1 are shown here.

For the analysis the same fitting approach was chosen as for the on-ground calibration analysis described in Kleipool et al. (2018): all measurements are processed up to and including the sun-distance correction, divided by the reference frame from the following orbit and transformed to an azimuth–elevation super pixel grid of size $10 \times 10$ pixels. In Fig. 7 the normalized

measurements of such a super pixel (row 20, column 30) is shown for each detector for QVD1. There is a substantial, but smooth variation in the azimuth direction of about 15 % between -10° and +10°.

In the elevation direction the variation between -4° and +4° is small, but the drop in signal for larger deviations is sudden. To derive the relative irradiance key data, a fit is performed on this super pixel grid using an 8th order Chebyshev polynomial both in the azimuth and elevation direction. The polynomial was fitted to values between -10° and +10° in the azimuth direction and

-4° and +4° in the elevation direction as shown in Fig. 8. The higher angular sampling shows more detail and is best reflected with a polynomial higher in order than used for the on-ground data.

The residuals that remain after application of the Chebyshev fit as shown in Fig. 9 are largely caused by the variation between the orbits, see also Section 12. Every track along the azimuth/elevation has a distinct amplitude. The origin of this variation is not yet exactly known. This random variation that is around $1-3 \times 10^{-3}$ poses a lower bound of the exactness of the fit for

the available data. To validate the integration of processor and key data, double processing is performed: data that has already been corrected with the derived CKD is re-analysed for remaining effects. Double processing irradiance data with the derived relative irradiance CKD reduces the standard deviation to the order of $\times 10^{-4}$. This result is an order of magnitude better than what was achieved with double processing of the CKD derived from on-ground calibration data.





Calibrating the solar diffusers in-flight by moving the platform proved to be very successful: apart from the better accuracy, the new CKDs also show more detail than the on-ground CKDs. In Fig. 10 the CKDs from on-ground and in-flight calibration are shown with their difference which is up to 2 percent points. With version 2 of the L1b processor accurate key data for both internal diffusers will be available.

## 12 Absolute radiometry and instrument degradation

In-flight the instrument is exposed to UV light and cosmic radiation potentially causing degradation of optical and electronic parts. Apart from the degradation, electronic drifts can occur that lower the radiometric accuracy of the radiance and irradiance. During nominal operations calibration and monitoring measurements are scheduled on a regular basis to be able to correct for degradation and drift effects. The TROPOMI instrument is designed such that all optical elements in the Earth view mode are included in the optical path when the Sun is measured. Thereby all degradation occurring in the spectrometers should cancel out when the reflectance is considered. To be able to determine the Earth's reflectance, the instrument measures the Sun via internal quasi volume diffusers (QVDs) on a regular basis. The main diffuser (QVD1) is used every day and once a fortnight and the backup diffuser (QVD2) every week during nominal operation.

Although in reflectance degradation effects should cancel, this only holds if the solar port degradation is corrected for. In addition, it is highly desirable to isolate the degradation of the spectrometers and correct for it separately. In this way irradiance, radiance and reflectance are all stand-alone products.

Thus, the challenge is to separate the various degradation and drift effects and identify where exactly in the instrument they occur. For diagnostics the internal light sources and solar measurements can be used. The internal light sources show a decrease in output as described in Section 4 and are therefore not as useful as the irradiance measurements. Radiance measurements in general show too much variability in themselves and would require too much input from atmospheric models to be useful for the derivation of an independent and sufficiently accurate degradation correction.

During the commissioning phase of TROPOMI several effects were identified: the degradation of the diffusers (QVD1 and QVD2) used for irradiance measurements, a gradual spectrally dependent increase of the throughput in the UV spectrometer and a drift of the CCD gain for the UVN spectrometers. With the exception of the UV spectrometer, so far no degradation could be identified within the other spectrometers. If – in the future – also degradation can be identified for other spectrometers than the UV, the L1b processor has the capability to correct spectrometer degradation for all bands provided that calibration key data can be derived.

To describe the spectrometer and solar port degradation for both internal diffusers QVD1 and QVD2, a model is used where the different contributions multiply to the total observed signal. For each (illuminated) detector pixel the total degradation $D_{\mathrm{tot}}$ is described by a linear system per QVD:

$$
\begin{aligned}
D_{\mathrm{tot},q1}(k) &= D_{q1}(t_{q1}(k)) \cdot D_{\mathrm{com}}(k) \cdot D_{\mathrm{spec}}(k) \cdot R_k \\
D_{\mathrm{tot},q2}(k) &= D_{q2}(t_{q2}(k)) \cdot D_{\mathrm{com}}(k) \cdot D_{\mathrm{spec}}(k) \cdot R_k \cdot P_k
\end{aligned}
\tag{1}
$$





The total degradation is composed by a contribution from the specific diffuser $D_{q1}$ or $D_{q2}$, a contribution which is common
for both diffusers $D_{\text{com}}$, a contribution which can be attributed specifically to the spectrometer $D_{\text{spec}}$ and the residuals $R_k$ and
$P_k$. The residuals describe mainly measurement to measurement variations, part of them are common to both diffusers ($R_k$)
and some are specific for QVD2 ($P_k$). The variable $k$ denotes the time in orbit numbers. The specific degradation curves $D_{q1}$
and $D_{q2}$ are perfect exponential curves, where the decay rate for $D_{q2}$ is about six times smaller than for $D_{q1}$, the ratio of usage
between QVD1 and QVD2. The component $D_{\text{com}}$ denotes an exponential decay which is observed for irradiance measurements
both via QVD1 and QVD2 and cannot be explained by the difference in usage. This common degradation could have its cause
in the folding mirror, which is part of the irradiance path for both diffusers, the telescope or within the spectrometers.

To solve the linear system in Eq. (1), the solar irradiance measurements for QVD1 and QVD2 are collected. Only the frames
at the solar reference angle at $1.269°$ azimuth and $0°$ elevation are used. Used are the weekly irradiance measurements for
QVD2 and for QVD1 only the ones which are taken on the same day as the QVD2 measurements. The total usage time
of the two QVDs $t_{q1}(k)$ and $t_{q2}(k)$ is extracted from the in-flight calibration database and is used to determine the ratio
in degradation rate. After various corrections, such as electronic gain (and gain drift for UVN) and Earth–Sun distance, the
images for all spectrometers are re-gridded on their respective wavelength grid to remove the spectral smile. The images are
then divided by the reference image (orbit 2818 for QVD1 and orbit 2819 for QVD2) and re-gridded onto a coarser grid of
super pixels to reduce noise. For each of these super pixels the linear system in Eq. (1) is solved. For the UVIS, NIR and SWIR
no spectrometer degradation $D_{\text{spec}}$ could be determined and this term is therefore set to unity. The solutions for $D_{q1}$, $D_{q2}$ and
$D_{\text{com}}$ are all three exponential decay functions and perfectly smooth in the temporal dimension. All temporal measurement to
measurement variation is contained in the residual images $R_k$ and $P_k$.

The UV spectrometer has a spectral overlap with the UVIS in the range 312–330 nm. In this spectral range the degradation
should be identical for UV and UVIS if the degradation is occurring within the optical path they have in common, so diffusers,
folding mirror and telescope. By extrapolating the common degradation $D_{\text{com}}$ derived for UVIS into the UV spectral range,
the spectrometer specific degradation $D_{\text{spec}}$ for UV can be isolated. Figure 11 shows the resulting modelled UV spectrometer
degradation and the actual measured signal ratios on the left side. On the right hand side can be seen that the ratio of signals as
measured by UV and UVIS at 317 nm evolves smoothly once the residual temporal variations ($R_k$ and $P_k$) are corrected for.
By using spatial and spectral filtering and some averaging in time the solutions for $D_{q1}$, $D_{q2}$, $D_{\text{com}}$ and $D_{\text{spec}}$ are turned into
unbinned calibration key data for each spectrometer and QVD.

Figures 12–14 show for each UVN spectrometer irradiance signals from several orbits before and after correction with the
new degradation key data. For the latest orbit the correction is based on extrapolation within the L1b processor. In this example
the extrapolation is over about 3.5 months. The residuals after correction are smaller than 0.1 %. The degradation is highest for
short wavelengths in the UV (Fig. 12) and UVIS (Fig. 13), is low in the NIR (Fig. 14) and negligible for SWIR as visible in
Fig. 15.

In the UV, the spectrometer specific degradation $D_{\text{spec}}$ shows a characteristic spectral signature where the signal increases
over time. In the left part of Fig. 12 it can be seen that the spectrometer ageing is stronger than the diffuser degradation and
negates the effect of the latter. The UVIS in Fig. 13 shows a clear spectral dependence but no increase in signal with time.



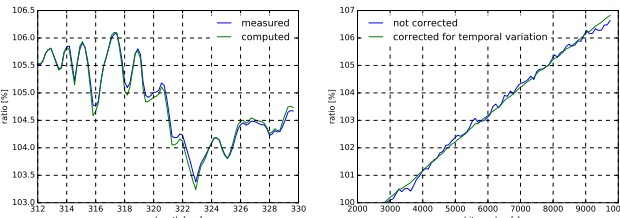

**Figure 11.** Left: Spectral degradation in the UV spectrometer from measured irradiance ratios between band 2 and band 3 (blue) and $D_{\mathrm{spec}}$ as computed from the model (green) for orbit 8849. The other model contributions are not included here. Right: Evolution of the ratio of measured signals at 317 nm, with (green) and without (blue) correction of the temporal variation ratios. Note the constant rate of increase of 1 % per 1000 orbits.

For the NIR (Fig. 14) the measurement to measurement variations are larger than the degradation. Figure 15 shows that no wavelength dependence of the degradation can be detected for the SWIR spectrometer. This is not unexpected considering the small covered wavelength range (90 nm) and the absolute wavelength scale (2400 nm). The observed change in irradiance signal shows measurement to measurement variations, in the model in Eq. 1 these are the residuals $R_k$ and $P_k$ and they are shown in Fig. 16. These temporal variations are spectrally and spatially smooth for each spectrometer and non-deterministic.

There is a close correlation between the temporal variations of UVIS and NIR and the variations observed with UV and SWIR. The two pairs are not correlated and the UV-SWIR variations have about half the magnitude of of the UVIS-NIR variations. The residuals are not corrected for in the L1b processor. In the UV, UVIS and NIR the derived degradation keydata has the same character, only the amounts differ. For SWIR the spread of signal values from measurement to measurement is large compared to the average change over time, this can be clearly seen in Fig. 17. The signal spread in SWIR seems to be dominated by

electronic noise and not by irradiance measurement variations. The observed degradation in SWIR is qualitatively not similar to the UVN degradation. A neutral degradation CKD will therefore be used for SWIR.

    As a baseline for L1b processing the degradation is defined relative to the start of the E2 phase, this is orbit 2818 for QVD1 and orbit 2819 for QVD2. The corrections to the absolute irradiance calibration as described in Section 13 is tied to the same orbits, in this way all corrections are consistent. As degradation continues with time, the calibration key data will need regular

updates to ensure that the accuracy is not lowered due to extrapolation of the key data in the L1b processor. In Table 4 the degradation per 1000 orbits is shown per band and for the different contributions. For the UV spectral degradation at 317 nm the increase in signal amounts to almost 1 % per 1000 orbits.

## 13    Absolute irradiance calibration

The absolute irradiance calibration aims to ensure that the sensitivity of the TROPOMI instrument for each measured wave-

length (i.e. at each detector pixel) is adjusted such that the measured irradiance reflects the solar output per wavelength. This was done during the on-ground calibration campaign (OCAL), but there were several issues with the stimuli as reported in





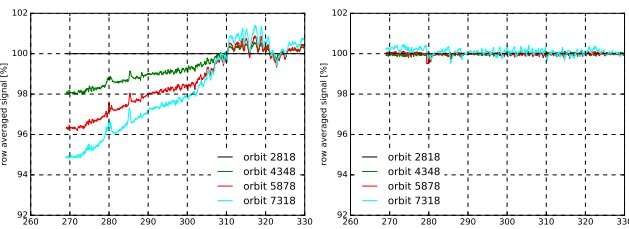

**Figure 12.** Row averaged irradiance signal of the UV detector via QVD1 for orbits 2818 (black flat line), 4348 (green), 5878 (red) and 7318 (cyan). The plot on the left is without degradation correction and clearly shows the spectral dependence of the degradation and the increase in signal for some spectral ranges. The right plot shows the corrected signal where the degradation CKD used only the measurements up to and including orbit 5878. The latest orbit in the plot (cyan) is corrected using extrapolation in the L1b processor.

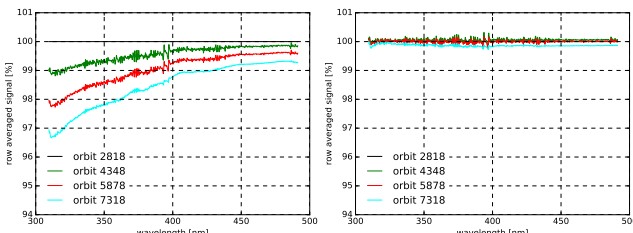

**Figure 13.** Row averaged irradiance signal of the UVIS detector via QVD1 for orbits 2818 (black flat line), 4348 (green), 5878 (red) and 7318 (cyan). The plot on the left is without degradation correction and clearly shows the spectral dependence of the degradation but no increase in signal as the UV detector in Fig. 12. The right plot shows the corrected signal where the degradation CKD used only the measurements up to and including orbit 5878. The latest orbit in the plot (cyan) is corrected using extrapolation in the L1b processor.

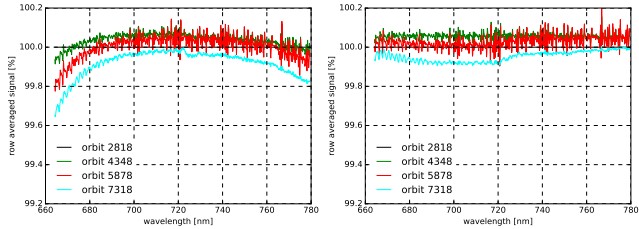

**Figure 14.** Row averaged irradiance signal of the NIR detector via QVD1 for orbits 2818 (black flat line), 4348 (green), 5878 (red) and 7318 (cyan). The plot on the left is without degradation correction and shows much less degradation than in the UV and UVIS. The right plot shows the corrected signal where the degradation CKD used only the measurements up to and including orbit 5878. The latest orbit in the plot (cyan) is corrected using extrapolation in the L1b processor.




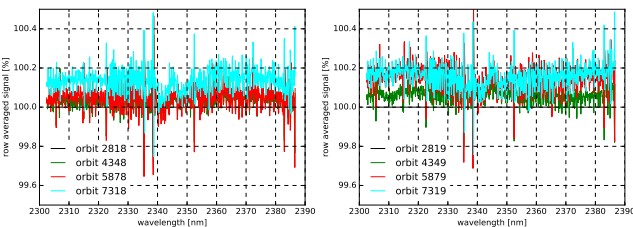

**Figure 15.** Normalized irradiance measurements for the SWIR detector for QVD1 (left) and QVD2 (right) over time. Signals are shown as row-averages versus wavelength. Shown are orbits 2818/2819 (black flat line), 4348/4349 (green) 5878/5879 (red) and 7318/7319 (cyan). The spread of signal values for the different orbits does neither show a trend nor a wavelength dependence.

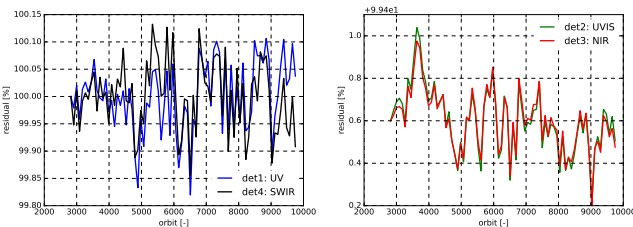

**Figure 16.** The residual temporal variation components $R_k$ of the irradiance measurements as modelled in Eq. (1) over time. The residuals have been computed independently for all spectrometers and the similarity is not imposed by the model. Left panel: UV (blue) and SWIR (black). The magnitude is notably smaller than for UVIS (green) and NIR (red) in the right panel.

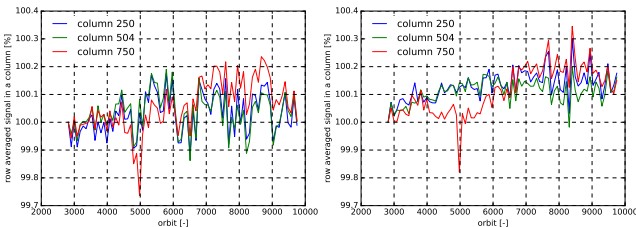

**Figure 17.** Normalized irradiance measurements for the SWIR detector for QVD1 (left) and QVD2 (right). Signals are shown as row-averages for three columns. The spread of signal values is large compared to the average change over time.




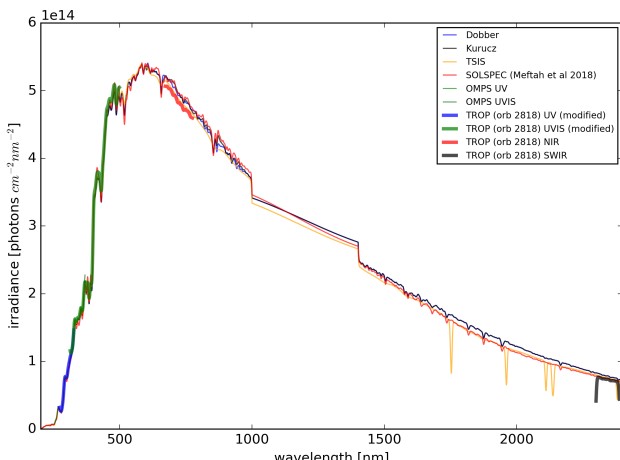

**Figure 18.** The solar spectrum according to reference spectra (thin lines) such as Dobber (red), Kurucz (black), TSIS (orange), SOLSPEC (red), and OMPS (green) and the four TROPOMI spectrometers via diffuser QVD1 (thick lines). The spectra were all convoluted with a Gaussian with a 3.0 nm standard deviation on the Dobber grid.

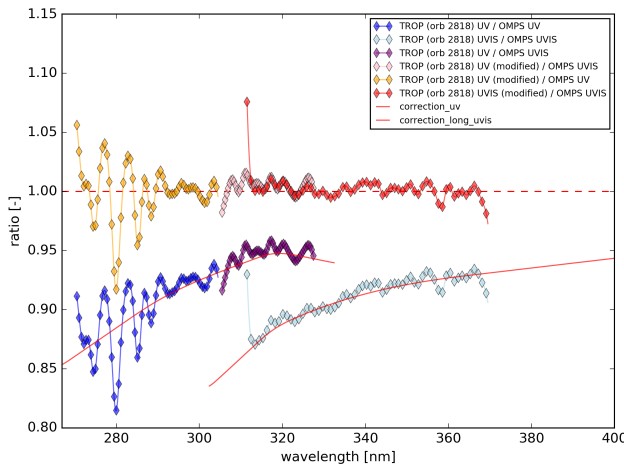

**Figure 19.** The TROPOMI UV and UVIS spectra via QVD1 (blue, purple and light blue), divided by the OMPS spectrum. The ratio differs between bands 2 (blue) and 3 (light blue) in the spectral overlap region of TROPOMI. In orange, pink and red the modified spectra are shown. They were modified with the cubic spline, indicated by the red lines. All spectra are convolved with a 1.0 nm Gaussian kernel.





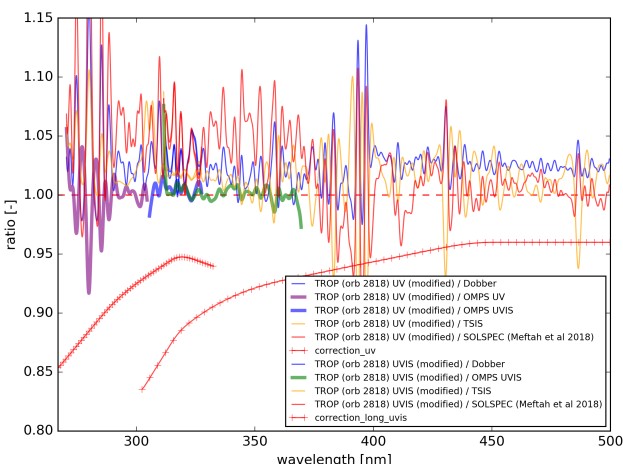

**Figure 20.** The ratio of the corrected spectrum via QVD1 with respect to the various reference spectra, convolved with a $\sigma = 1.0\,\mathrm{nm}$ Gaussian. The thick purple, blue and green lines indicate the ratio with OMPS data where the correction is based on. The correction splines (red crosses) bring the TROPOMI spectrum in good general agreement with several reference spectra. The spectrum shows similarity to the other spectra on a larger scale, except for a bias in the order of 0–5 %.

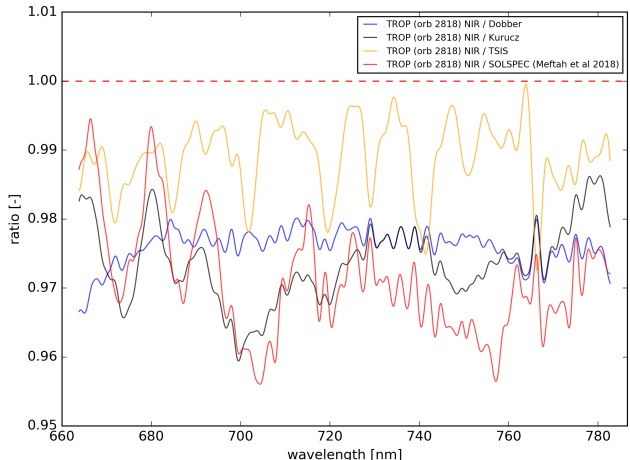

**Figure 21.** The ratio of the NIR spectrum via QVD1 with respect to the Dobber spectrum (blue), Kurucz (green), TSIS (orange) and SOLSPEC (red). All NIR spectra are convolved with a $\sigma = 1.0\,\mathrm{nm}$ Gaussian kernel.





**Table 4.** The degradation per band per 1000 orbits as determined up to orbit 9748. Shown is the modelled degradation of the QVD1 ($D_{q1}$) and the QVD2 ($D_{q2}$). together with their common degradation $D_{\mathrm{com}}$. Moreover, in the UV detector a spectral ageing component $D_{\mathrm{spec}}$ of up to 1% per 1000 orbits reverses the apparent degradation. The maximum and standard deviation of the residuals $\max(R_k)$ and $\sigma(R_k)$ in UV (and SWIR) are consistently lower than those in UVIS and NIR. The modelled degradation for NIR and SWIR is in the order of the maximum residual value. The values for SWIR have *not* been implemented in a L1b correction.

| Per 1000 orbits | QVD1 & common | QVD2 & common | common | spectrometer | max. residual | std. residual |
|---|---|---|---|---|---|---|
| | $D_{q1} \cdot D_{\mathrm{com}}$ | $D_{q2} \cdot D_{\mathrm{com}}$ | $D_{\mathrm{com}}$ | $D_{\mathrm{spec}}$ | $\max(R_k)$ | $\sigma(R_k)$ |
| **Band 1** | 0.979 | 0.546 | 0.465 | -0.189 | 0.036 | 0.014 |
| **Band 2** | 0.717 | 0.385 | 0.323 | -0.701 | 0.025 | 0.010 |
| **Band 3** | 0.485 | 0.249 | 0.205 | - | 0.065 | 0.022 |
| **Band 4** | 0.203 | 0.101 | 0.082 | - | 0.062 | 0.022 |
| **Band 5** | 0.036 | 0.011 | 0.007 | - | 0.059 | 0.020 |
| **Band 6** | 0.034 | 0.011 | 0.007 | - | 0.057 | 0.020 |
| **Band 2 : 317nm** | 0.693 | 0.373 | 0.313 | -0.995 | - | - |
| **Band 3 : 317nm** | 0.699 | 0.383 | 0.325 | - | - | - |
| **Band 7** | *-0.012* | *-0.033* | *-0.037* | - | 0.022 | 0.009 |
| **Band 8** | *-0.016* | *-0.031* | *-0.033* | - | 0.021 | 0.009 |

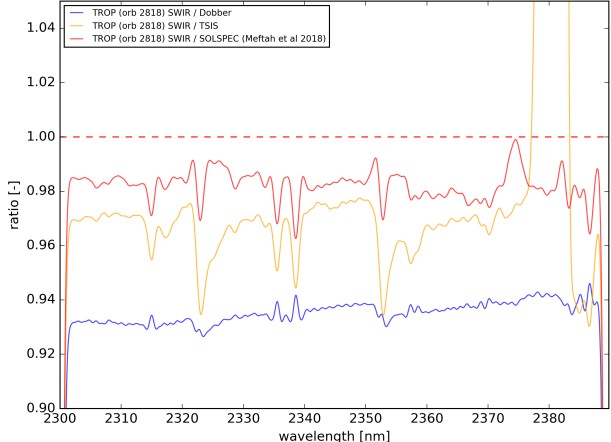

**Figure 22.** The ratio of the SWIR spectrum via QVD1 with respect to the Dobber spectrum (blue), the TSIS (orange), and SOLSPEC (red). The TSIS spectrum deviates more than 15 % around 2380 nm, so it is off the scale. All SWIR spectra are convolved with a $\sigma = 0.5$ nm Gaussian kernel.

Kleipool et al. (2018). Especially in bands 1–3 the calibration measurements were affected by a low signal-to-noise ratio. In-flight measurements revealed that the absolute irradiance calibration for UV and UVIS is inconsistent. Band 2 of the UV





spectrometer and band 3 of the UVIS spectrometer have some spectral overlap, and correctly calibrated data should give the
same irradiance values for both bands in this wavelength range. As can be seen in Fig. 19 from the uncorrected data, this is not
the case. With only the on-ground calibration applied, the irradiance in the UV is visibly lower than that of other instruments
and there is a discontinuity between the UV (270–330 nm) and UVIS (310–500 nm) in their overlap region. To remove this
inconsistency for UV and UVIS, the solar spectrum of TROPOMI is compared to different published solar reference datasets
as shown in Fig. 18.

A well-known solar reference is the high resolution Dobber spectrum ($\pm 0.014$ nm per pixel) (Dobber et al., 2008) and the
Kurucz spectrum (Chance and Kurucz, 2010), which are high resolution composites of different solar measurement campaigns.
It covers the spectral range of the TROPOMI instrument, but especially in the UV range it is unclear if it is reliable. Other
datasets are from the TSIS instrument on board of the International Space Station (LASP Interactive Solar Irradiance Datacen-
ter, 2019), and the SIM (Woods et al., 2009) and SOLSPEC data (Thuillier et al., 2003). The SIM spectrum was once corrected
(Woods et al., 2009) with a bias to be in closer agreement to the Dobber (Dobber et al., 2008) and Thuillier SOLSPEC spectra
(Thuillier et al., 2003), but more recently the spectrum has been published in its original, uncorrected state (Harder et al., 2010),
showing much more resemblance to the SOLSPEC spectrum published by Meftah et al. (2018) and the TROPOMI spectrum.
Other instruments that measure solar spectra in the TROPOMI UV and UVIS spectral range are the Ozone Monitoring Instru-
ment (OMI) and the Ozone Mapping and Profiler Suite (OMPS). The former has been calibrated using the Dobber spectrum. To
reduce possible interdependencies by using a composite spectrum, we have chosen to use the independently calibrated OMPS
solar irradiance spectrum (Jaross et al., 2014) as a reference level for the absolute calibration of the TROPOMI irradiance
spectrum in the spectral range of bands 1–3. The OMPS instrument has very similar spectral characteristics as TROPOMI and
the published spectrum is solely based on OMPS data. The difference between the TROPOMI and OMPS spectrum can be
largely resolved by multiplying the TROPOMI spectrum with a (piecewise) linear function.

The on-ground spectral calibration for the TROPOMI instrument was done using a FEL lamp, which has a spectrally smooth
output suggesting that the calibration did not introduce spectral features. Speckle introduced by the internal diffusers was
filtered as described in Kleipool et al. (2018). Therefore, any adjustment of the absolute calibration should also be spectrally
smooth. Each spectrum is convolved switch a Gaussian kernel, with a standard deviation that is representative for the effective
spectral resolution of the instrument or larger.

The TROPOMI solar spectral irradiance in UV and UVIS is adjusted by finding piecewise linear approximations of the ratio
of TROPOMI and OMPS, and joining them with a cubic spline. The initial ratio and the corrected ratio together with the used
cubic spline are shown in Fig. 19 for UV and the UVIS band 3. Band 4 of the UVIS spectrometer is outside the spectral range
or OMPS. In Fig. 20 all the corrected UV and UVIS bands are shown relative to reference data. For validation and clarity, the
data shown is convolved with a kernel with standard deviation of 1 nm, while the fit of the cubic splines used for the correction
is based on convolutions with a kernel with standard deviation of 3 nm to reduce the impact of spectral lines. It can be seen
that the spectra of the UV and UVIS spectrometers have been modified by at least 5 % and at most 15 %. The gap between UV
and UVIS in the spectral overlap region has disappeared. For wavelengths above 450 nm the correction is a bias, bringing the


data in good agreement with SOLSPEC and TSIS. All the shown data is for diffuser QVD1, but the correction was derived for both diffusers and is very similar.

For the NIR and SWIR spectrometers the deviations from the reference spectra are much smaller than for the UV and also seem to consist mainly of a spectrally flat bias. As shown in Fig. 21 the spectrum of the NIR spectrometer is approximately 1.5–3.5 % lower than the reference spectra. The SWIR spectrum shown in Fig.22 is approximately 1.5–7 % lower than the reference spectra, but it is closest to the SOLSPEC spectrum published by Meftah et al. (2018) which resembles the SIM spectrum in its uncorrected state (Harder et al., 2010) (not shown). Considering the spread of the reference spectra and the uncertainty of

the TROPOMI on-ground calibration of around 1 % it seems unwise to change the TROPOMI NIR and SWIR solar spectra to match any of the other references. Therefore no modifications of the irradiance on-ground calibration was performed for the NIR and SWIR spectrometers and their calibration remains independent from other instruments and references.

## 14   Changes to the nominal operations baseline

Several of the findings from the commissioning phase resulted in changes to the planned nominal operations baseline. An

overview of the nominal operations baseline can be found in KNMI (2017). The main change is that the matching background measurements for the radiance measurements are performed with a closed folding mirror (FMM) to ensure that remaining light from the eclipse side of the Earth is blocked off. The FMM is a life limited item, so this is only performed in orbits where the FMM is employed anyway to perform calibration measurements. The different orbit types were re-arranged such that the radiance background is measured 6–7 times per day. All calibration and background measurements are scheduled in

full eclipse only. As as consequence some calibration measurements are occasionally performed inside the SAA and flagged as such. Another substantial change is that the irradiance measurements are performed close to the solar azimuth angle where the absolute calibration has been performed. To achieve this, the platform performs a slew manoeuvre with the on-board reaction wheels before the irradiance measurements as for the relative irradiance calibration described in Section 11. This reduces the angle range over which the relative irradiance correction needs to be applied and prepares for the possibility that the solar angle

moves outside the design range, which can happen if – for example at the end of the mission lifetime – the orbit is changed.

The radiance signals vary over latitude during each orbit. During the commissioning phase the instrument settings for the different signal levels were fine-tuned for optimal signal while minimizing saturation. Small changes were also applied to instrument settings to measure irradiance and the internal light sources. All insights from the commissioning phase were already included in an updated nominal operations scenario before the start of the nominal operations (E2) phase in orbit 2818

on 30 April 2018.

The only change in nominal operations since the start of the E2 phase was the reduction of the radiance co-addition time from 1080 ms to 840 ms starting in orbit 9388 on 6 August 2019. This results in a shorter minimal along-track sampling distance: before it was approximately 7.1 km at nadir and it is now about 5.6 km. In across-track direction the minimal sampling distance at nadir is around 3.6 km for bands 2–6, about 7.2 km for bands 7–8 and around 28.8 km for band 1. The lower limits for the





bands are due to different row binning values for UVN (bands 1–6) and the spectrometer's instantaneous field of view (SWIR, bands 7–8).

## 15 Conclusions

The TROPOMI instrument on-board the Sentinel-5 Precursor satellite is functioning very well. The thermal and orbital stability is very good. Only during orbital manoeuvres instrument temperatures can increase, impacting mainly the spectral calibration

of the SWIR spectrometer. Thermal instabilities will be flagged in the updated L1b processor. The internal light sources WLS, CLED and DLED show a continuous decrease in output of at most 0.9 % per thousand orbits. The CCD output gain of the UVN detectors displays drifts. Based on regular performed calibration measurements with internal sources, these drifts can be corrected within better than 0.1 %. High signals can lead to pixel saturation and charge blooming for the UVN detectors. This occurs mainly in bands 4 and 6. The v2 of the L1b processor version includes a new algorithm where affected pixels are

flagged. The validation of the geolocation showed that an additional yaw-angle correction 0.002 radians was needed to allow for changes due to gravity release after launch. This correction had already been implemented in v1 of the L1b processor and is active since the beginning of the nominal operations phase. Small corrections were also derived for the spectral annotation of the UVIS and SWIR spectrometers. In the UV spectrometer a slit irregularity was observed after launch. The drop in signal for several rows is corrected in the processor update.

The calibration of the solar angle dependence of the irradiance radiometry which was too inaccurate on-ground, was successfully performed in-flight by moving the platform to cover the different angles. The resulting key data has a higher sampling and a higher accuracy than what was previously available.

In-flight several degradation effects have been observed, they are strongest in the UV spectral range and can be isolated and modelled. They will be corrected with the version 2 of the L1b processor using time-dependent calibration key data. Calibration

key data for instrument properties which change over time, such as the diffuser degradation or the UVN gain drift, now have a time axis. The updated processor is also able to handle possible future degradation effects both for irradiance and radiance data, the algorithms are in place for all detectors.

During in-flight commissioning some inconsistencies of the on-ground calibration results were found and corrections were developed. This concerns mainly the absolute irradiance radiometry calibration. From comparison with several reference solar

irradiance spectra, a spectrally smooth correction was applied to the calibration of UV and UVIS.

The version 2 of the L1b processor with all updated and new key data presented in this paper, is planned to be in operation from 2020 on.

*Data availability.* The plots and analysis presented in this article contain modified Copernicus Sentinel-5 Precursor data [2017–2019]. The S5P user products are available via https://scihub.copernicus.eu/. In-flight monitoring and calibration data can be found via http://www.

tropomi.eu/data-products/level-1.



*Author contributions.* RB performed data analysis and investigated the degradation in the UV spectrometer. QK is the instrument scientist and project lead of the L1b data processing and calibration development. RL was in charge of data processing chain and developed the transient flagging algorithm. JL developed all geometric calibration analysis software and is responsible for the geolocation annotation in the L1b data processor. AL is the optical expert and planned the in-flight commissioning and calibration activities and programmed the instrument settings. EL is the mathematical consultant and was responsible for all algorithm definitions, and he analysed and reported on most electronic calibrations and developed the degradation corrections. PM was responsible for all database engineering required for the calibration processing. EvdP derived the relative radiometric response of the irradiance, which also included the detectors' PRNU, and corrections to the absolute irradiance. NR is system architect and acting lead of the L1b data processing development team and he has developed the blooming correction. FV was system engineer for the overall software development and responsible for the release management. PV is acting principal investigator for the TROPOMI payload on-board the Sentinel-5 Precursor satellite.

*Competing interests.* The authors declare that they have no conflict of interest.

*Acknowledgements.* The work presented in this paper was funded by NSO and ESA.





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
