# Peer review of "In-flight calibration results of the TROPOMI payload on-board the Sentinel-5 Precursor satellite"

_Atmospheric Measurement Techniques, 2019_

## Referee Comment (RC1) · Anonymous Referee #3 · 2 Mar 2020

Section 5: The text states that in-flight linearity deviates from on-ground by no more than 1%. This seems rather large. Is this a statistically significant deviation? This deserves more discussion.

Section 7: This is an important topic, but the authors choose to devote only a short qualitative discussion to it. It would be helpful to the reader to provide some idea of the errors involved. At what error level does the flagging occur?

Section 8: The authors state they have only addressed geolocation in Bands 4-7. Geolocation in the shorter bands, esp. Band 3, are also important and validation should be possible except for Band 1. The authors should at least discuss what their plans are

to validate these bands.

Section 9: A similar comment about wavelength registration. The authors imply there is no source of wavelength information other than from L2 products and there are no products providing this information for Bands 1 & 2. Yet the spectral registration in these bands is no less important than at longer wavelengths. The authors can at least acknowledge the problem and discuss their plans to deal with it.

Section 11: The discussion in this section (esp. the paragraph starting at line 260) was somewhat confusing. The authors should consider two alternatives to remedy this: provide a bit more explanation to the reader, or eliminate some of the details that are the source of the confusion. I recommend the latter because it's not clear what is to be learned from these details.

Section 12: In Line 285 the authors seem to throw cold water on any technique, other than on-board calibrations, to derive or validate radiometric change. It is quite reasonable that the authors have not had a chance to implement any of the well-documented techniques for validating the calibration, but they should refrain from suggesting these were omitted because they lack useful information.

I think I follow the 'competing change' argument described in Lines 330-335, but I doubt most readers will. The authors need to describe explicitly what about Figures 12 & 13 indicates increasing detector response competing with diffuser degradation.

Table 3: These numbers appear to be in percent. The authors should say so explicitly.

Section 13: The authors imply at the start of Section 12 that the reflectance calibration of TropOMI is an important quantity, but they fail to address its accuracy. If that is outside the purview of this paper, the authors should say so. The authors also fail to discuss in this section the effect that adjusting the irradiance calibration has on measured Earth TOA reflectances. Since the radiance calibration wasn't mentioned, the reader is left to assume that all the adjustments described in Section 13 are being

applied in inverse to the instrument's reflectance calibration. What is the justification for doing so? The authors provide no insight as to why the pre-launch irradiance calibration might be so much in error. How do they know that the radiance calibrations are not in error by an equal or nearly equal amount?

Grammar comment: Use of the word "for" in connection with "corrected" should be accompanied by an object rather than a subject. "We correct for something" rather than "Something is corrected for."

---

## Referee Comment (RC2) · Anonymous Referee #2 · 5 Mar 2020

The paper gives a detailed overview of the in-flight calibration of the TROPOMI instrument on-board the Sentinel-5P satellite platform, launched by the European Space Agency (ESA) as part of the Copernicus Earth observation programme by the European Union. The results from this work leads to an update of the operation Level-1b processor to the version 2, expected to be operational end of 2020.

The paper is well written and in general gives a consistent overview about the investigated and improved calibration issues. The paper will be an important reference for all users of version 2 of the TROPOMI level 1b product.

Main concern from my side is the foreseen radiometric re-calibration of the irradiance

and its impact on the reflectance, see detailed comments below.

With the recommended changes, this paper should be published in AMT.

Detailed comments:

*Figures:*
Fig 1-3,5,6,11-22: The used fonts are to small, enlarge.
Fig. 7-9: at least the numbers at the color scales need to be larger.
For not so young eyes, the numbers and labels in the printed paper are difficult to read.
Please change.

*Abstract:*
p1 l13: "processing from 2020 on". From my knowledge, late 2020 is the current foreseen start for the version 2 L1b processor. Please adapt the date, also in the conclusions.

*1 Introduction:*

p2 l41: please define the term "orbit types" (probably the measurement sequence along the orbit.)

p3: Please add (at the end of the introduction) a sketch of the instrument design, it would be very useful for the following paragraphs: Where is the calibration unit, what are the light paths, where are the diffuser etc. Please also add a paragraph or a sketch to the detector layout: row and column is frequently used in the text, but nowhere the spatial and spectral direction is explicitly stated.

*2 Thermal stability:*

It is observed, that the thermal stability is reduced after orbital manoeuvres. Is there a reason or at least an educated guess for this behaviour? If yes, please add.

*7 Pixel saturation and charge blooming:*

Are there estimations available, how frequently saturation and charge blooming occur? Which are the suspect conditions (snow? tropical clouds? something else?). Please add.

*8 Geolocation:*

p9 l161: ..or along-rack.. -> or along-track

*9 Spectral annotation:*

It is stated, that the calibration key data for the wavelength calibration are updated according to the wavelength fits in the Level 2 algorithms. Are the key data directly used as wavelength axis? There is no Level 1 wavelength calibration?

*10 Slit irregularity:*

Especially for this section, the definition of rows/columns versus spatial/spectral direction in the introduction would be very useful!

Figure 6:
Change y-axis name to 'binned row counter' (this is the used name in the text). Please add in the caption, that the row 335-337 corresponds to the binned counter 144.

*12 Absolute radiometry and instrument degradation*

p16 l303ff:
"The specific degradation curves ... are perfect exponential curves".
Here it is assumed, that the degradation behaves exponential, so write something like: "It is assumed, that the diffuser degradation behaves exponential with time. Therefore, the specific degradation curves ... are modelled as exponential curves". Also the exponential behaviour of $D_{com}$ is an assumption and should be stated as such.

p16 l315ff:
"For each of these super pixels the linear system in Eq. (1) is solved. For the UVIS, NIR and SWIR no spectrometer degradation $D_{spec}$ could be determined and this term

is therefore set to unity."
I think, this is the wrong order: For UVIS, NIR, and SWIR, no spectrometer degradation can be derived, therefore $D_{spec}$ is set to 1.0 for theses channels. With this assumption, the linear equations system is solved for each super pixel if UVIS, NIR and SWIR. Right?
Please also give a number, how many pixels are in one super-pixel.

p 16 l317:
"The solutions for $D_{q1}$ , $D_{q2}$ and $D_{com}$ are all three exponential decay functions and perfectly smooth in the temporal dimension."
Your model fits exponential decay functions for this quantities, therefore this is trivial message. What could be stated here is something like:
The assumption of an exponential decay for $D_{q1}$, $D_{q2}$ and $D_{com}$ is approved by the small residuals $R_k/P_k$, as shown by the right plots in Fig 12-14.

The explanation for estimating $D_{spec}$ for the UV leaves a few questions open:

$D_{com}$ is extrapolated to the UV region. What about $D_{q1}/D_{q2}$? What type of extrapolation do you use, so what are the assumptions made? Towards shorter wavelengths, the degradation is expected to increase. According to the left plot in Fig. 11, this is not the case for $D_{spec}$.

p 17, l 347-352:
For the forward processing, an extrapolation of the degradation parameters is used. It is stated, that this new degradation parameters will be regularly updated by incorporating the recent measurements. With the update, also the extrapolation will change. This might introduce jumps in the irradiance time series, which might be an issue for users. Is there a strategy to monitor and/or avoid this? Please add some information about the details here.

p22, Table 4: The *mean* degradation per Band is given, right? Please clarify.

[Figure]

*13 Absolute irradiance calibration:*

Why is the OMPS irradiance measurement choosen as the reference measurement for the radiometric calibration? To my knowledge, OMPS is an unusual solar reference measurement. OMPS does not even distribute there irradiance measurements as regular product. The cited literature [Jaross 2014] gives no information about the absolute radiometric calibration except a plot together with an unnamed "synthetic" spectrum. If possible, at a reference for the radiometric calibration of the OMPS irradiance.

Recently re-calibrated and published solar spectra are SOLSPEC (Meftah et al 2018) or SCIAMACHY (Hilbig et al, 2018), which would be a better choice. Both are also independent from other reference spectra.

Nothing is said about the radiance calibration. Is the discontinuity observed in the overlap region also visible in radiances? What about the reflectance?

The light path is the same for radiance and irradiance except the QVD. The QVD is the same for the the UV / UVIS overlap and cannot cause the discontinuity. Therefore, in the reflectance the discontinuity should cancel. If only the irradiance is mitigated here, the discontinuity is introduced in the reflectance. The radiance calibration and the impact of the irradiance mitigation on the reflectance needs to be discussed here.

*Conclusions:*

p 25, l 434/ 436: 'v1' / 'v2' change to 'version 1' / 'version 2'.
p 25, l 449 : radiometry -> radiometric

*References:*

Many references contain both the the DOI based URL and a direct URL. Only the DOI URL as permanent URL is needed, skip the second URL (which is also not added consistently...).

For Ingmann et al. the URL is erroneous.

---

## Referee Comment (RC3) · Ruediger Lang (Referee) · 6 Mar 2020

The paper by Ludewig et al. on the In-flight calibration results of the TROPOMI payload on-board the Sentinel-5 Precursor satellite is an important contribution to the meanwhile significant history of knowledge in characterising this class of instruments, since the launch of GOME-1 on ERS-2. The paper is well written and organized and it follows the individual calibration steps in a systematic way. This is important in order to be able to compare the presented results and calibration approaches to past instrument in-flight calibration efforts, and to compare their associated calibrated level-1 radiance product quality performance. This may then inform both, ongoing reprocessing efforts for

the derivation of fundamental climate data-records from instruments like SCIAMACHY, GOME-2, OMI, OMPS, as well as the preparation of future instrument on-ground calibration and commissioning campaigns, and their continuous like Sentinel-4/5, TEMPO, but also the High-Priority candidate Copernicus CO2 monitoring Mission (CO2M).

I can recommend the paper for publication in AMT but would like the authors first to consider and address the following issues (accompanied by a list of minor remarks) Irradiometric calibration, observed degradation and its correction.

Section 12 and 13 describe the approach taken to correct for some partially significant, observed degradation effects especially in the UV. The overall approach seems sound (section 12). However it is not obvious for me how the degradation model approach and application in section 12 is related, or better decoupled, from the correction of the observed, partially quite significant offsets (up to 15%) in the absolute irradiance calibration of the solar port (section 13).

My understanding from the paper is that the derived spectrometer component (from the 312 to 330 nm region) has been accounted for by a degradation correction, which is, again to my understanding, applied spectrally neutral to the full UV detector irradiance. Is this correction then also applied for Earthshine measurements, as one would expect it to be, because it is considered an effect of the common optical path? In case yes, I guess that the normalization day/orbit 2818/2819 is then used for an adjustment to OMPS, such that any likely degradation happening to the irradiance signals until this point is corrected for by reference to OMPS. Again, one expects an unknown degradation to have happened also to the Earthshine path until orbit 2818/2819, which would then lead to a differential degradation in reflectance after adjustment of the solar irradiance, and especially in case nothing is done additionally for the Earthshine data (and probably there are also some finite yet different accuracies for the radiometric key-data to be taken into account).

The choice of OMPS seems also very subjective. While it is stated that OMPS irradiance has been "independently calibrated", it is not stated what "independently" would mean in this context (without adjustment to reference spectra? If yes, then this should be stated). I would maintain that it remains just a choice. The results show a close to 3% difference with the Dobber et al. spectrum after adjustment. In contrast, all three GOME-2 instruments shave shown smaller residuals than 3% to the Dobber reference spectrum, above 300 nm at the beginning of live, without (!) adjustment (so using the on-ground derived key-data only). So this choice of a reference solar spectrum would leave a potential unknown "offset" of 2 to 3% with respect to other instruments and their absolute calibration after degradation correction. Since 2 to 3% accuracy is effectively the current limit on the knowledge of the solar irradiance accuracy in the UV and VIS wavelength region in general, such a choice for sure can be made, but it should be presented as the limit of the knowledge in the absolute calibration accuracy then also for this mission. Moreover, this would then also be the limit of knowledge on the Earthshine radiance accuracy, with a potentially even larger error on the reflectance.

In this respect, the question is why an independent Earthshine degradation modelling has been ruled out. For previous missions GOME-1, 2 and SCIAMACHY degradation modelling using global averages of cloud free Earthshine data showed quite some success, and also Libyan desert degradation modelling should not be ruled out.

Finally, the derived spectrometer component in Section 12 seems to be on the order of 1% per 1000 orbits (Figure 11). In contrast the observed WLS and LED signal degradations seem to be lower or on the same order. I am wondering why the use of the internal light sources then have been ruled out for degradation monitoring or even correction, or how their "output degradation" could have been identified as such, when the identified spectrometer component is on the same order or even more significant. Is there an optical component in the path (like another folding mirror) between the spectrometer and the WLS, such that any direct Earthshine degradation modelling using these sources cannot easily be done? It might be interesting to look at the ratio of calibrated SMR and calibrated WLS, and their (differential) evolution over time and

spectrally in this context.

Additional comments

Section 3, l 80ff: How exactly non-linear is the observed decrease of the light sources and can this decrease be attributed to the sources or is it already part of the optical chain for WLS? It should not be ruled out that this is simply a consequence of the spectrometer degradation observed in Section 12 (see before).

Section 5, l.90ff: I would assume the temperature dependency of the dark current has been measured on-ground. From these measurements it could be stated here what is the projected dark current orbital dependency using the observed orbital detector thermal stability from HKTM.

L.110ff: The change in the gain during manoeuvres is not further explained. Can any reason be given for this?

l. 145ff: It would be interesting (and helpful for future missions) to get an idea (statistically) on the extend of blooming in pixel space. E.g. by providing a histogram (or table) on the number of occurrences over the number of pixels affected per event. Does such a statistic exist?

Section 8 on geo-referencing: Has any attempt be made for geo-rectification using VIIRS data? This should provide very accurate geo-referencing knowledge also on the point-spread function. Can anything be said about the alignment of the other bands not used in the geo-referencing analysis? Or can some qualitative assumption be derived from the optical setup (telescope) and alignment? A discussion would be needed here I think.

Section 10 on slit irregularities: From Figure 6 it looks like the WLS exhibits significant spectral structure. Why is this? Actually, wouldn't a highly structured spectrum like the solar lead to a better correction?

Section 11 on goniometry: The azimuthal maximum variation of the sun should be

reported in this Section in order to motivate/justify the restriction to 10 degrees, even though 15 degrees have been measured. Is the orbit stabilized, and for how long in the mission? Or in other words, is there any restriction in future ground track drifts concerning the validity range of this data?

Section 11, on the origin of the remaining residuals in the goniometry key-data derived in-flight: I would guess that they are probably a combination of diffuser features, speckles, and especially instrument drifts between individual measurements and the temporal position of the normalization measurement. In addition, one should find the pattern of the observed degradation correction residual in such a potential drift, I would assume. Since the measurement period was quite long (400 orbits), and it was in an early state of the mission, can effects like gain drifts during this period, and as reported in the earlier sections, be ruled out? It would be good to discuss the status of the mission at the time of the dedicated measurement period (start orbit, overall platform thermal stability etc...), and if the measurements have been filtered for outliers.

Section 12, degradation model: Why would one expect that all components are "perfectly exponential". At least in the long-run. Since this is not what is observed with other instruments, and for sure not in case of a potential mirror contribution. Is there a long-term trend observed in the Rk and Pk components?

l. 364: "but especially in the UV range it is unclear if it is reliable": Which spectrum is referred here to? Since we have observed that the Dobber et al., spectrum shows clearly better results for GOME-2 for wavelength below 300 nm at the beginning of the mission and without any adjustments than all other available reference spectra. I fear that at this stage this is no discussion about the truth, but probably more about inter-instrument consistencies.

Editorial comments

General: Although it has been describe multiple times elsewhere, a table of band numbering associated with source region "UV", "UVIS, "NIR" "SWIR" and associated wavelength ranges would be of help for the reader to have at hand up-front. Since band numbers, detector labels and source regions are used multiple times in exchangeable ways in the paper.

Figure 5: "...within the requirements" -> add black lines in brackets p. 10ff: The plots in Figure 6 and the reported row numbers in the text (e.g. line 208) are different. The caption indicates the Figure shows the binned count. Somewhere at least a written translation should be made. E.g. in the caption: bin x corresponds to pixels yy. or similar.

p14, ;l263: Check sentence: "For double processing, so (?) ..." l.380 switch -> with

---

## Author Comment (AC2)

**Review comment amt-2019-488-RC2**

**Reviewer: Anonymous Referee #2**

**Dear referee,**

Thank you for your detailed review of our article. Our responses to your remarks, questions and considerations can be found in the table below. The performed changes to the manuscript are listed in the Section "Detailed Changes".

**Response**

| Item         | Referee comment                                                                    | Author's response                                               |
|--------------|------------------------------------------------------------------------------------|-----------------------------------------------------------------|
| Fig 1-       | The used fonts are to small, enlarge                                               | The plots have been adapted or enlarged. Captions have          |
| 3,5,6,11-22  |                                                                                    | been adapted accordingly.                                       |
| Fig. 7-9     | At least the numbers at the color scales need to be larger. For not so young       | The plots have been adapted or enlarged. Captions have          |
|              | eyes, the numbers and labels in the printed paper are difficult to read. Please    | been adapted accordingly.                                       |
|              | change.                                                                            |                                                                 |
| p1  13,      | "processing from 2020 on". From my knowledge, late 2020 is the current             | Yes, that is correct. Currently the planning is for late |
| abstract     | foreseen start for the version 2 L1b processor. Please adapt the date, also in     | 2020. Adapted both occurrences.                                 |
|              | the conclusions.                                                                   |                                                                 |
| p2 l41,      | please define the term "orbit types" (probably the measurement sequence            | Adapted.                                                        |
| introduction | along the orbit.)                                                                  |                                                                 |
| РЗ,          | Please add (at the end of the introduction) a sketch of the instrument design, it  | Added a new figure with the functional schematic of             |
| introduction | would be very useful for the following paragraphs: Where is the calibration        | TROPOMI. The spatial and spectral direction is now              |
|              | unit, what are the light paths, where are the diffuser etc. Please also add a      | added to the text several times.                                |
|              | paragraph or a sketch to the detector layout: row and column is frequently         |                                                                 |
|              | used in the text, but nowhere the spatial and spectral direction is explicitly     |                                                                 |
|              | stated.                                                                            |                                                                 |
| Section 2    | It is observed, that the thermal stability is reduced after orbital manoeuvres. Is | Added. The thermal stability is reduced when the                |
| Thermal      | there a reason or at least an educated guess for this behaviour? If yes, please    | pointing of the radiant cooler is not optimal as can be         |
| stability    | add.                                                                               | the case during manoeuvres.                                     |

| ltem          | Referee comment                                                                         | Author's response                                           |  |
|---------------|-----------------------------------------------------------------------------------------|-------------------------------------------------------------|--|
| Section 7     | Are there estimations available, how frequently saturation and charge                   | Information on the pixel saturation has been added. For     |  |
| Pixel         | blooming occur? Which are the suspect conditions (snow? tropical clouds?                | the blooming we do not have detailed statistics yet as      |  |
| saturation,   | something else?). Please add.                                                           | the new version is not in use yet.                          |  |
| blooming      |                                                                                         |                                                             |  |
| Section 8     | p9 l161:or along-rack> or along-track                                                   | Adapted.                                                    |  |
| Geolocation   |                                                                                         |                                                             |  |
| Section 9     | It is stated, that the calibration key data for the wavelength calibration are          | Currently there is no online Level 1 wavelength             |  |
| Spectral      | updated according to the wavelength fits in the Level 2 algorithms. Are the key         | calibration. The L1 key data is based on on-ground          |  |
| annotation    | data directly used as wavelength axis? There is no Level 1 wavelength                   | calibration and adapted with the in-flight insights. L2     |  |
|               | calibration?                                                                            | retrieval algorithms perform their own wavelength           |  |
|               |                                                                                         | fitting where the L1 wavelength assignment is used as a     |  |
|               |                                                                                         | starting point.                                             |  |
| Section 10    | Especially for this section, the definition of rows/columns versus                      | Adapted.                                                    |  |
| Slit          | spatial/spectral direction in the introduction would be very useful!                    |                                                             |  |
| irregularity  |                                                                                         |                                                             |  |
| Figure 6      | Change y-axis name to 'binned row counter' (this is the used name in the text).         | This has already been adapted following the initial         |  |
|               | Please add in the caption, that the row 335-337 corresponds to the binned               | review comments.                                            |  |
|               | counter 144.                                                                            |                                                             |  |
| p 16   303 ff | "The specific degradation curves are perfect exponential curves". Here it is            | It was found that the exponential fits resulted in a better |  |
| Section 12    | assumed, that the degradation behaves exponential, so write something like:             | fit than other functions. We made this clearer in the text  |  |
| absolute      | "It is assumed, that the diffuser degradation behaves exponential with time.            | that this is the model.                                     |  |
| radiometry    | Therefore, the specific degradation curves are modelled as exponential                  |                                                             |  |
| and           | curves". Also the exponential behaviour of D com is an assumption and should |                                                             |  |
| instrument    | be stated as such.                                                                      |                                                             |  |
| degradation   |                                                                                         |                                                             |  |
| P 16   315 ff | "For each of these super pixels the linear system in Eq. (1) is solved. For the         | For UVIS, NIR and SWIR no spectrometer degradation          |  |
| Section 12    | UVIS, NIR and SWIR no spectrometer degradation D spec could be determined    | was found so the term was set to 1.                         |  |
| absolute      | and this term is therefore set to unity." I think, this is the wrong order: For         |                                                             |  |
| radiometry    | UVIS, NIR, and SWIR, no spectrometer degradation can be derived, therefore              | Added the size of the super pixels and some more            |  |
| and           | D spec is set to 1.0 for theses channels. With this assumption, the linear   | explanation on it.                                          |  |
| instrument    | equations system is solved for each super pixel if UVIS, NIR and SWIR. Right?           |                                                             |  |
| degradation   | Please also give a number, how many pixels are in one super-pixel.                      |                                                             |  |

| Item         | Referee comment                                                                              | Author's response                                         |
|--------------|----------------------------------------------------------------------------------------------|-----------------------------------------------------------|
| p 16   317   | "The solutions for Dq1 , Dq2 and Dcom are all three exponential decay functions              | The phrasing has been changed to make clear that the      |
| Section 12   | and perfectly smooth in the temporal dimension." Your model fits exponential                 | exponential decay is the model.                           |
| absolute     | decay functions for this quantities, therefore this is trivial message. What                 |                                                           |
| radiometry   | could be stated here is something like: The assumption of an exponential                     | For the ratio of Dq1/q2 the assumption is made, that the  |
| and          | decay for Dq1, Dq2 and Dcom is approved by the small residuals Rk/Pk, as shown               | degradation is exposure based, so we use the total time   |
| instrument   | by the right plots in Fig 12-14. The explanation for estimating D spec for the UV | of usage as input. The spectral ageing Dspec of the UV    |
| degradation  | leaves a few questions open: Dcom is extrapolated to the UV region. What                     | spectrometer does indeed behave different than the        |
|              | about Dq1/Dq2? What type of extrapolation do you use, so what are the                        | diffuser degradation. The signal increases with time.     |
|              | assumptions made? Towards shorter wavelengths, the degradation is                            |                                                           |
|              | expected to increase. According to the left plot in Fig. 11, this is not the case            |                                                           |
|              | for D spec .                                                                      |                                                           |
| P 17,   347- | For the forward processing, an extrapolation of the degradation parameters is                | Added remark on jumps in the data.                        |
| 352          | used. It is stated, that this new degradation parameters will be regularly                   |                                                           |
|              | updated by incorporating the recent measurements. With the update, also the                  |                                                           |
|              | extrapolation will change. This might introduce jumps in the irradiance time                 |                                                           |
|              | series, which might be an issue for users. Is there a strategy to monitor and/or             |                                                           |
|              | avoid this? Please add some information about the details here.                              |                                                           |
| P 22, Table  | The *mean* degradation per Band is given, right? Please clarify.                             | Yes for the bands/ wavelength it's the mean               |
| 4            |                                                                                              | degradation. Rephrased the caption.                       |
| 13 Absolute  | Why is the OMPS irradiance measurement choosen as the reference                              | The OMPS solar irradiance is not distributed as a         |
| irradiance   | measurement for the radiometric calibration? To my knowledge, OMPS is an                     | separate product, but is part of every L1b file, we added |
| calibration: | unusual solar reference measurement. OMPS does not even distribute there                     | additional references.                                    |
|              | irradiance measurements as regular product. The cited literature [Jaross 2014]               | We now added explicitly that the reflectance is changing, |
|              | gives no information about the absolute radiometric calibration except a plot                | and explained the observed inconsistencies from on-       |
|              | together with an unnamed "synthetic" spectrum. If possible, at a reference for               | ground calibration.                                       |
|              | the radiometric calibration of the OMPS irradiance. Recently re-calibrated and               | The OMPS spectrum was chosen for several reasons: it      |
|              | published solar spectra are SOLSPEC (Meftah et al 2018) or SCIAMACHY (Hilbig                 | has similar instrument characteristics, it is an active   |
|              | et al, 2018), which would be a better choice. Both are also independent from                 | mission and a single instrument spectrum and not a        |
|              | other reference spectra. Nothing is said about the radiance calibration. Is the              | composite spectrum. This point is made clearer in the     |
|              | discontinuity observed in the overlap region also visible in radiances? What                 | text now.                                                 |
|              | about the reflectance? The light path is the same for radiance and irradiance                | As shown in the paper we compared the results to          |
|              | except the QVD. The QVD is the same for the the UV / UVIS overlap and cannot                 | different references.                                     |

| Item         | Referee comment                                                                   | Author's response |
|--------------|-----------------------------------------------------------------------------------|-------------------|
|              | cause the discontinuity. Therefore, in the reflectance the discontinuity should   |                   |
|              | cancel. If only the irradiance is mitigated here, the discontinuity is introduced |                   |
|              | in the reflectance. The radiance calibration and the impact of the irradiance     |                   |
|              | mitigation on the reflectance needs to be discussed here.                         |                   |
| Conclusions  | 'v1' / 'v2' change to 'version 1' / 'version 2', radiometry -> radiometric        | Adapted.          |
| p 25, l 434/ |                                                                                   |                   |
| 436:         |                                                                                   |                   |
| p 25, l 449  |                                                                                   |                   |
| References   | Many references contain both the the DOI based URL and a direct URL. Only         | Corrected.        |
|              | the DOI URL as permanent URL is needed, skip the second URL (which is also        |                   |
|              | not added consistently).                                                          |                   |
| References   | For Ingmann et al. The URL is erroneous                                           | Corrected.        |

**Detailed changes**

**List of changes to version 2**

The page and line numbering in the Table below is according to version 2 which was public on the discussion page. The comments on the version 1 (the one which was initially sent out to the reviewers) have already been included in version 2.

[revised manuscript text omitted]

**Changes to initial version 1**

The changes below have been performed to the initial version 1 sent out to the referees. These changes were already included in the version 2 which was published on the discussion page and are listed below for completeness.

| Line number Fig/Table
(version1/version2) | Original (version 1)                  | Update (version 2)                                                                          |
|----------------------------------------------|---------------------------------------|---------------------------------------------------------------------------------------------|
| /Table 1                                     |                                       | Added table on main characteristics.                                                        |
| 24/24                                        |                                       | Added " The main characteristics of TROPOMI are listed in Table 1. "                        |
| 23/23                                        | 5.5km x 3.5km                         | Put non-rounded number to be consistent with new table: 5.6km x 3.6km                       |
| 420/423                                      | "before it was approximately 7 km     | Put non-rounded number to be consistent with new table: "before it was                      |
|                                              | at nadir and it is now about 5.5 km.  | approximately 7.1km at nadir and it is now about 5.6km. In across-track direction the       |
|                                              | In across-track direction the minimal | minimal sampling distance                                                                   |
|                                              | sampling distance                     | at nadir is around 3.6km for bands 2–6, about 7.2km for bands 7–8 and around 28.8km         |
|                                              | at nadir is around 3.5km for bands    | for band 1."                                                                                |
|                                              | 2–6, about 7km for bands 7–8 and      |                                                                                             |
|                                              | around 28km for band 1."              |                                                                                             |
| Caption Fig.5/Fig.5                          | "The differences for low and high     | Added "(black lines)":                                                                      |
|                                              | row numbers are now mostly within     | "The differences for low and high row numbers are now mostly within the                     |
|                                              | the requirements and more             | requirements (black lines) and more                                                         |
|                                              | symmetrical."                         | symmetrical."                                                                               |
| 208                                          | 335337                                | Changed to em-dash: "335–337"                                                               |
| Caption Fig.6/Fig.6                          | "Note that the row numbering is       | Changed to "Note that the binned row count is shown in the plots, the affected              |
|                                              | showing the binned count."            | detector rows are rows 335337."                                                             |
| 220/221                                      |                                       | Added: "Detector rows 335 and 336 correspond in this example to the binned row              |
|                                              |                                       | counter 144."                                                                               |
| 263/265                                      | "For double processing, so re-        | Re-phrased to: "To validate the integration of processor and key data, double               |
|                                              | analysing data that is corrected with | processing is performed: data that has already been corrected with the derived CKD is       |
|                                              | the derived relative irradiance CKD,  | re-analysed for remaining effects. Double processing irradiance data with the derived       |
|                                              | the standard deviation reduces to     | relative irradiance CKD reduces the standard deviation to the order of $	imes$ 10 –4 . This |

| Line number Fig/Table | Original (version 1)                   | Update (version 2)                                                                   |
|-----------------------|----------------------------------------|--------------------------------------------------------------------------------------|
| (version1/version2)   |                                        |                                                                                      |
|                       | the order of ×10 −4 , this is an order | result is an order of magnitude better than what was achieved with double processing |
|                       | of magnitude better than what was      | of the CKD derived from on-ground calibration data."                                 |
|                       | achieved with the on-ground data."     |                                                                                      |

---

## Author Comment (AC3)

**Review comment amt-2019-488-RC3**

**Reviewer: Rüdiger Lang**

**Dear referee,**

Thank you for your detailed review of our article. Our responses to your remarks, questions and considerations can be found in the table below. The performed changes to the manuscript are listed in the Section "Detailed Changes".

**Response**

| Item             | Referee comment                                                                                                                                                                                                                                                                                                                                                                                                                                                                                                                                                                                                                                                                                                                                                                                                                                                                                                                                                                                                                                                                                                                                                                                                                                                                                                                                                                                                                                                    | Author's response                                                                                                                                                                                                                                                                                                                                                                                                                                                                                                                                                                                                                                                                                                                                                                                                                                                                                                      |  |
|------------------|--------------------------------------------------------------------------------------------------------------------------------------------------------------------------------------------------------------------------------------------------------------------------------------------------------------------------------------------------------------------------------------------------------------------------------------------------------------------------------------------------------------------------------------------------------------------------------------------------------------------------------------------------------------------------------------------------------------------------------------------------------------------------------------------------------------------------------------------------------------------------------------------------------------------------------------------------------------------------------------------------------------------------------------------------------------------------------------------------------------------------------------------------------------------------------------------------------------------------------------------------------------------------------------------------------------------------------------------------------------------------------------------------------------------------------------------------------------------|------------------------------------------------------------------------------------------------------------------------------------------------------------------------------------------------------------------------------------------------------------------------------------------------------------------------------------------------------------------------------------------------------------------------------------------------------------------------------------------------------------------------------------------------------------------------------------------------------------------------------------------------------------------------------------------------------------------------------------------------------------------------------------------------------------------------------------------------------------------------------------------------------------------------|--|
| Section
12/13 | Section 12 and 13 describe the approach taken to correct for some partially significant, observed degradation effects especially in the UV. The overall approach seems sound (section 12). However it is not obvious for me how the degradation model approach and application in section 12 is related, or better decoupled, from the correction of the observed, partially quite significant offsets (up to 15%) in the absolute irradiance calibration of the solar port (section 13). My understanding from the paper is that the derived spectrometer component (from the 312 to 330 nm region) has been accounted for by a degradation correction, which is, again to my understanding, applied spectrally neutral to the full UV detector irradiance. Is this correction then also applied for Earthshine measurements, as one would expect it to be, because it is considered an effect of the common optical path? In case yes, I guess that the normalization day/orbit 2818/2819 is then used for an adjustment to OMPS, such that any likely degradation happening to the irradiance signals until this point is corrected for by reference to OMPS. Again, one expects an unknown degradation to have happened also to the Earthshine path until orbit 2818/2819, which would then lead to a differential degradation in reflectance after adjustment of the solar irradiance, and especially in case nothing is done additionally for the Earthshine | From the on-ground calibration we are sure that the irradiance calibration is not correct for bands 1-3 (made this clearer in the text). For the on-ground calibration of the radiance there is no such evidence. By setting the reference for the (spectrally smooth) diffuser degradation and the absolute irradiance adaptions on the same orbit, the diffuser degradation up to that point is taken into account. The spectral ageing in the UV spectrometer is corrected in radiance and irradiance for the entire mission. The spectral features already present in orbit 2818 are therefore removed and the smooth correction of the absolute irradiance takes care of the diffuser degradation. We made this point clearer in the text. If there is a remaining inconsistency in radiance this needs to be addressed in future validation for example via Earth targets, this has also been added to the text. |  |

| Item       | Referee comment                                                                       | Author's response                                       |
|------------|---------------------------------------------------------------------------------------|---------------------------------------------------------|
|            | data (and probably there are also some finite yet different accuracies for the        |                                                         |
|            | radiometric key-data to be taken into account).                                       |                                                         |
| Section    | The choice of OMPS seems also very subjective. While it is stated that OMPS           | We made clearer in the text why OMPS was chosen.        |
| 12/13      | irradiance has been "independently calibrated", it is not stated what                 | The idea was to be able to relate the changes to a      |
|            | "independently" would mean in this context (without adjustment to reference           | single instrument and not a composite spectrum.         |
|            | spectra? If yes, then this should be stated). I would maintain that it remains just a | And indeed, eventually it is a choice.                  |
|            | choice. The results show a close to 3% difference with the Dobber et al. spectrum     | We added a clarification on validation using Earth      |
|            | after adjustment. In contrast, all three GOME-2 instruments shave shown smaller       | targets, this is future work. Therefore we are also not |
|            | residuals than 3% to the Dobber reference spectrum, above 300 nm at the               | presenting any updated numbers for the radiance         |
|            | beginning of live, without (!) adjustment (so using the on-ground derived key-data    | and reflectance accuracy yet.                           |
|            | only). So this choice of a reference solar spectrum would leave a potential           |                                                         |
|            | unknown "offset" of 2 to 3% with respect to other instruments and their absolute      |                                                         |
|            | calibration after degradation correction. Since 2 to 3% accuracy is effectively the   |                                                         |
|            | current limit on the knowledge of the solar irradiance accuracy in the UV and VIS     |                                                         |
|            | wavelength region in general, such a choice for sure can be made, but it should be    |                                                         |
|            | presented as the limit of the knowledge in the absolute calibration accuracy then     |                                                         |
|            | also for this mission. Moreover, this would then also be the limit of knowledge on    |                                                         |
|            | the Earthshine radiance accuracy, with a potentially even larger error on the         |                                                         |
|            | reflectance. In this respect, the question is why an independent Earthshine           |                                                         |
|            | degradation modelling has been ruled out. For previous missions GOME-1, 2 and         |                                                         |
|            | SCIAMACHY degradation modelling using global averages of cloud free Earthshine        |                                                         |
|            | data showed quite some success, and also Libyan desert degradation modelling          |                                                         |
|            | should not be ruled out.                                                              |                                                         |
| Section 12 | Finally, the derived spectrometer component in Section 12 seems to be on the          | Analysing WLS data has given valuable insight on the    |
|            | order of 1% per 1000 orbits (Figure 11). In contrast the observed WLS and LED         | spectral ageing in the UV and the spectral overlap      |
|            | signal degradations seem to be lower or on the same order. I am wondering why         | with UVIS. The light path of the WLS includes           |
|            | the use of the internal light sources then have been ruled out for degradation        | additional optics and is not identical to the Sun or    |
|            | monitoring or even correction, or how their "output degradation" could have           | Earth path. When using WLS for calibration purposes     |
|            | been identified as such, when the identified spectrometer component is on the         | WLS features could be introduced into the L1            |
|            | same order or even more significant. Is there an optical component in the path        | radiance/irradiance.                                    |
|            | (like another folding mirror) between the spectrometer and the WLS, such that         |                                                         |
|            | any direct Earthshine degradation modelling using these sources cannot easily be      |                                                         |

| ltem           | Referee comment                                                                       | Author's response                                      |  |
|----------------|---------------------------------------------------------------------------------------|--------------------------------------------------------|--|
|                | done? It might be interesting to look at the ratio of calibrated SMR and calibrated   | Added an explanation to the difference in light paths  |  |
|                | WLS, and their (differential) evolution over time and spectrally in this context.     | for DLED and WLS                                       |  |
| Section 3, I   | How exactly non-linear is the observed decrease of the light sources and can this     | The decrease in signal can be caused by the source     |  |
| 80ff           | decrease be attributed to the sources or is it already part of the optical chain for  | itself, the source's specific optical path and the     |  |
|                | WLS? It should not be ruled out that this is simply a consequence of the              | instrument. Clarified this in the text.                |  |
|                | spectrometer degradation observed in Section 12 (see before).                         |                                                        |  |
| Section 5,     | I would assume the temperature dependency of the dark current has been                | The temperature dependency of the dark current has     |  |
| 1.90ff         | measured on-ground. From these measurements it could be stated here what is           | not been measured on-ground.                           |  |
|                | the projected dark current orbital dependency using the observed orbital detector     |                                                        |  |
|                | thermal stability from HKTM.                                                          |                                                        |  |
| L.110ff:       | The change in the gain during manoeuvres is not further explained. Can any            | So far no reason has been found to explain this        |  |
|                | reason be given for this?                                                             | behaviour. All available housekeeping parameters       |  |
|                |                                                                                       | have been checked but no correlation was found.        |  |
| l. 145ff       | It would be interesting (and helpful for future missions) to get an idea              | We have added numbers for the occurrence of pixel      |  |
|                | (statistically) on the extend of blooming in pixel space. E.g. by providing a         | saturation. For the blooming itself a full statistical |  |
|                | histogram (or table) on the number of occurrences over the number of pixels           | analysis is possible once version 2 of the L01b        |  |
|                | affected per event. Does such a statistic exist?                                      | processor is active.                                   |  |
| Section 8 on   | Has any attempt be made for geo-rectification using VIIRS data? This should           | We have not attempted any cross-validation of the      |  |
| geo-           | provide very accurate geo-referencing knowledge also on the point-spread              | geolocation with VIIRS or other satellites.            |  |
| referencing    | function. Can anything be said about the alignment of the other bands not used in     | Considering the limited spatial resolution of          |  |
|                | the geo-referencing analysis? Or can some qualitative assumption be derived from      | TROPOMI we don't think that a comparison to higher     |  |
|                | the optical setup (telescope) and alignment? A discussion would be needed here I      | resolution instruments would have added to the         |  |
|                | think.                                                                                | results. A discussion on qualitative assumptions for   |  |
|                |                                                                                       | the other bands has been added.                        |  |
| Section 10     | From Figure 6 it looks like the WLS exhibits significant spectral structure. Why is   | Figure 6 is an irradiance image showing characteristic |  |
| on slit        | this? Actually, wouldn't a highly structured spectrum like the solar lead to a better | spectral lines. The data has been corrected with key   |  |
| irregularities | correction?                                                                           | data derived from WLS data.                            |  |
| Section 11     | The azimuthal maximum variation of the sun should be reported in this Section in      | The natural solar azimuth range during solar           |  |
| on             | order to motivate/justify the restriction to 10 degrees, even though 15 degrees       | calibration measurements over one year is between -    |  |
| goniometry     | have been measured. Is the orbit stabilized, and for how long in the mission? Or in   | 10° and +6.0° (range is 16°) for an ANX MLST of        |  |
|                |                                                                                       | 13:30, which is the current mission requirement. The   |  |

| Item                    | Referee comment                                                                                                                                                                                                                                                                                                                                                                                                                                                                                                                                                                                                                                                                                                                                                                                                                                                                             | Author's response                                                                                                                                                                                                                                                                                                                                                                                                                                                                                                                                                                                                                                                                                                                       |
|-------------------------|---------------------------------------------------------------------------------------------------------------------------------------------------------------------------------------------------------------------------------------------------------------------------------------------------------------------------------------------------------------------------------------------------------------------------------------------------------------------------------------------------------------------------------------------------------------------------------------------------------------------------------------------------------------------------------------------------------------------------------------------------------------------------------------------------------------------------------------------------------------------------------------------|-----------------------------------------------------------------------------------------------------------------------------------------------------------------------------------------------------------------------------------------------------------------------------------------------------------------------------------------------------------------------------------------------------------------------------------------------------------------------------------------------------------------------------------------------------------------------------------------------------------------------------------------------------------------------------------------------------------------------------------------|
|                         | other words, is there any restriction in future ground track drifts concerning the validity range of this data?                                                                                                                                                                                                                                                                                                                                                                                                                                                                                                                                                                                                                                                                                                                                                                             | instrument requirements were therefore set to allow
for measurements in the range -10° to +10° in
azimuth and -4.25° to +4.25° in elevation. In reality
the solar baffle allows light for a larger range, the
measurements were therefore performed at the
largest possible range achievable with the platform.
The orbit is stabilized and follows Suomi NPP with a 3
to 5min delay. For nominal operations the solar port
should not run out of its calibrated range.
Furthermore the irradiance measurements are
performed around a fixed azimuth angle using the
reaction wheels. This is explained in Section 14, we
added a reference to this and a sentence on the
natural azimuth range. |
| Section 11              | Section 11, on the origin of the remaining residuals in the goniometry key-data derived in-flight: I would guess that they are probably a combination of diffuser features, speckles, and especially instrument drifts between individual measurements and the temporal position of the normalization measurement. In addition, one should find the pattern of the observed degradation correction residual in such a potential drift, I would assume. Since the measurement period was quite long (400 orbits), and it was in an early state of the mission, can effects like gain drifts during this period, and as reported in the earlier sections, be ruled out? It would be good to discuss the status of the mission at the time of the dedicated measurement period (start orbit, overall platform thermal stability etc), and if the measurements have been filtered for outliers. | The remaining residuals are - as you observe -
connected to the residuals observed in the
degradation correction. Added a clearer reference to
the residual discussion in Section 12. Thermal effects
can be excluded, the instrument was thermally
stabilized since very early in the commissioning
phase. The main part of electronic gain drifts is
corrected by the use of the normalization
measurements.
Added start orbit, remark on electronic drifts and in
Section 2 a remark on thermal stability for the
measurements.                                                                                                                                                                     |
| Section 12, degradation | Why would one expect that all components are "perfectly exponential". At least in the long-run. Since this is not what is observed with other instruments, and for                                                                                                                                                                                                                                                                                                                                                                                                                                                                                                                                                                                                                                                                                                                          | Until now an exponential function was found to be the best fit. If this changes in the future we can adapt                                                                                                                                                                                                                                                                                                                                                                                                                                                                                                                                                                                                                              |
| model:                  | sure not in case of a potential mirror contribution. Is there a long-term trend observed in the Rk and Pk components?                                                                                                                                                                                                                                                                                                                                                                                                                                                                                                                                                                                                                                                                                                                                                                       | it.
Added this remark to the text.                                                                                                                                                                                                                                                                                                                                                                                                                                                                                                                                                                                                                                                                                                   |
| l. 364                  | "but especially in the UV range it is unclear if it is reliable": Which spectrum is                                                                                                                                                                                                                                                                                                                                                                                                                                                                                                                                                                                                                                                                                                                                                                                                         | That is the difficulty with inter-instrument                                                                                                                                                                                                                                                                                                                                                                                                                                                                                                                                                                                                                                                                                            |
|                         | referred here to? Since we have observed that the Dobber et al., spectrum shows                                                                                                                                                                                                                                                                                                                                                                                                                                                                                                                                                                                                                                                                                                                                                                                                             | comparisons, in the end it is a matter of choice. We                                                                                                                                                                                                                                                                                                                                                                                                                                                                                                                                                                                                                                                                                    |

| Item        | Referee comment                                                                                                                                                                                                                                                                                                                                                     | Author's response                                                                                                                                                                                              |
|-------------|---------------------------------------------------------------------------------------------------------------------------------------------------------------------------------------------------------------------------------------------------------------------------------------------------------------------------------------------------------------------|----------------------------------------------------------------------------------------------------------------------------------------------------------------------------------------------------------------|
|             | clearly better results for GOME-2 for wavelength below 300 nm at the beginning
of the mission and without any adjustments than all other available reference
spectra. I fear that at this stage this is no discussion about the truth, but probably
more about inter-instrument consistencies.                                                             | tried to base our choice on how comparable the
spectral resolutions and ranges are and that the we
are traceable to a single instrument and not a
composite. This point was made clearer in the text. |
| General:    | Although it has been describe multiple times elsewhere, a table of band
numbering associated with source region "UV", "UVIS, "NIR" "SWIR" and
associated wavelength ranges would be of help for the reader to have at hand up-
front. Since band numbers, detector labels and source regions are used multiple
times in exchangeable ways in the paper. | This has already been adapted following the initial review comments.                                                                                                                                           |
| Figure 5/6  | "within the requirements" -> add black lines in brackets p. 10ff: The plots in
Figure 6 and the reported row numbers in the text (e.g. line 208) are different.
The caption indicates the Figure shows the binned count. Somewhere at least a
written translation should be made. E.g. in the caption: bin x corresponds to pixels
yy. Or similar.      | This has already been adapted following the initial review comments.                                                                                                                                           |
| p14, ;l263: | Check sentence: "For double processing, so (?)" I.380 switch -> with                                                                                                                                                                                                                                                                                                | This has already been adapted following the initial review comments.                                                                                                                                           |

**Detailed changes**

**List of changes to version 2**

The page and line numbering in the Table below is according to version 2 which was public on the discussion page. The comments on the version 1 (the one which was initially sent out to the reviewers) have already been included in version 2.

| Item          | Change                                                                                                                              |  |
|---------------|-------------------------------------------------------------------------------------------------------------------------------------|--|
| Nowfiguro     | Added new figure and caption at the beginning of the article. It shows a functional schematic of TROPOMI. Added a reference to this |  |
| New ligure    | figure in several places in the text.                                                                                               |  |
| Fig 1-3, 5-9, | Enlarged plots or adapted plots to increase fontsize and improve readability. Adapted captions and the references to the plots      |  |
| 11-22         | accordingly. For Fig.1 added "The triangles in the top panel show the gain ratio as derived from on-ground measurements."           |  |
| p1 13,        | Changed "processing from 2020 on" to "processing from late 2020 on"                                                                 |  |
| abstract      | Changed processing from 2020 on to processing from late 2020 on .                                                                   |  |

| Item                      | Change                                                                                                                                                                                                                                                                                                                                                                                                                                                                                                                                                                                                                                                                                                                                                                                                                                                                        |  |  |
|---------------------------|-------------------------------------------------------------------------------------------------------------------------------------------------------------------------------------------------------------------------------------------------------------------------------------------------------------------------------------------------------------------------------------------------------------------------------------------------------------------------------------------------------------------------------------------------------------------------------------------------------------------------------------------------------------------------------------------------------------------------------------------------------------------------------------------------------------------------------------------------------------------------------|--|--|
| p 1 l 22/23,
Table 1   | Adapted to be consistent with official PRF: 5.6-> 5.5, 7.2 ->7, 28.8->28                                                                                                                                                                                                                                                                                                                                                                                                                                                                                                                                                                                                                                                                                                                                                                                                      |  |  |
| p2   27 ff                | Replaced "The instrument is measuring the radiance on the day side of each orbit and once a day the irradiance via a dedicated solar
port as described in detail in KNMI (2017) and Kleipool et al. (2018)." By "The instrument is measuring the radiance on the day side of
each orbit and once a day the irradiance via a dedicated solar port as shown in Fig.1. Sun light passes through one of the two internal
quasi volume diffusers (QVD1 and QVD2) and is coupled via the folding mirror into the telescope of the instrument. A detailed
instrument description canbe found in KNMI (2017) and Kleipool et al. (2018)."                                                                                                                                                                                                                                 |  |  |
| p 2 l 41,
introduction | Changed "The timing and definition of the different orbit types was adapted to match the detected darkness of the eclipse." to "The timing and definition for the measurement sequences of the different orbit types was adapted to match the detected darkness of the eclipse."                                                                                                                                                                                                                                                                                                                                                                                                                                                                                                                                                                                              |  |  |
| p 3 l58                   | Added "All measurements described in this article were performed at the nominal temperatures with active thermal stabilization."                                                                                                                                                                                                                                                                                                                                                                                                                                                                                                                                                                                                                                                                                                                                              |  |  |
| p 3 l 62                  | Added "when the radiant cooler points in a sub-optimal direction"                                                                                                                                                                                                                                                                                                                                                                                                                                                                                                                                                                                                                                                                                                                                                                                                             |  |  |
| p 4 l 75 ff               | Replaced "output" by "observed signal"and "detector response".                                                                                                                                                                                                                                                                                                                                                                                                                                                                                                                                                                                                                                                                                                                                                                                                                |  |  |
| p 4 l 86                  | Added "Depending on the source and its location in the instrument, the listed values can contain contributions from degradation of the source, its specific optics, the diffusers, the folding mirror, the telescope and the spectrometers."                                                                                                                                                                                                                                                                                                                                                                                                                                                                                                                                                                                                                                  |  |  |
| p61114                    | Added "or other housekeeping parameters".                                                                                                                                                                                                                                                                                                                                                                                                                                                                                                                                                                                                                                                                                                                                                                                                                                     |  |  |
| p 7   138, 141            | Added " in the tropics". Changed "and" to "-". Added " In the tropics typically about 0.2-0.5% of the pixels are flagged for saturation in bands 4-6, other regions and bands are hardly ever affected."                                                                                                                                                                                                                                                                                                                                                                                                                                                                                                                                                                                                                                                                      |  |  |
| p7   140                  | Added: "For the CCD detectors spatial binning is applied: the charge of several successive detector rows is added in the register and then read out."                                                                                                                                                                                                                                                                                                                                                                                                                                                                                                                                                                                                                                                                                                                         |  |  |
| p 7 l141                  | Replace "this" by "the saturation issue"                                                                                                                                                                                                                                                                                                                                                                                                                                                                                                                                                                                                                                                                                                                                                                                                                                      |  |  |
| p 8 l143                  | Added:" (spatial direction)"                                                                                                                                                                                                                                                                                                                                                                                                                                                                                                                                                                                                                                                                                                                                                                                                                                                  |  |  |
| p 9 l157                  | Added: ", so only a narrow spectral range is available per UVN band."                                                                                                                                                                                                                                                                                                                                                                                                                                                                                                                                                                                                                                                                                                                                                                                                         |  |  |
| p 10 l 183ff              | Changed "For the SWIR and UVIS detectors the same effect is observed, so a mechanical change within the instrument during launch seems unlikely." to "For the SWIR, UVIS and NIR spectrometers the same effect is observed, so a mechanical change within the instrument itself during launch seems highly unlikely. For UV the signal to noise of the high resolution measurements with their small spectral range is too small to draw conclusions. The light for the UV and SWIR takes the same path up to and including the instrument slit and the UV spectrometer is part of the UVN optical bench as shown in Fig. 1. As the SWIR spectrometer shows the same effect as the UVIS and the NIR spectrometers and no difference is observed between UVIS and NIR, due to the instrument design it is highly unlikely that the UV spectrometer should behave differently." |  |  |
| p 10   188                | Added "A further validation is not foreseen, as the nominal radiance measurements have a larger groundpixel size."                                                                                                                                                                                                                                                                                                                                                                                                                                                                                                                                                                                                                                                                                                                                                            |  |  |

| Item         | Change                                                                                                                                      |  |  |
|--------------|---------------------------------------------------------------------------------------------------------------------------------------------|--|--|
| p 11   204   | Added "or data for other bands becomes available"                                                                                           |  |  |
| p 11   208   | Added "(spatial direction)"                                                                                                                 |  |  |
| p 11   210   | Changed "Therefore not the main instrument slit but the slit in the UV spectrometer is most likely causing the feature. "to "From the       |  |  |
|              | instrument design as shown in Fig. 1 it can be seen that not the main instrument slit but the slit in the UV spectrometer is most likely    |  |  |
|              | causing the feature."                                                                                                                       |  |  |
| p 11   214   | Added "as shown in Fig. 1"                                                                                                                  |  |  |
| p 11   217   | Added " (columns)"                                                                                                                          |  |  |
| p 12   236   | Changed "400 orbits" to " 400 consecutive orbits (starting in orbit 1247)"                                                                  |  |  |
| p 13   241   | Added " possible electronic drifts"                                                                                                         |  |  |
| n 14   261   | Added "The fitting window covers the natural yearly solar azimuth variation for the reference orbit with equator crossing time of 13:30     |  |  |
| p 14 1 201   | local solar time."                                                                                                                          |  |  |
| p 14   263   | Changed " see also Section 12" to "see also Section 12 for the description of the residuals"                                                |  |  |
| n 15   272   | Added "The slew manoeuvres are included in the nominal operations baseline as described in Section 14. This reduces the measured            |  |  |
| p151272      | azimuth range to less than ±1 ° around the reference angle."                                                                                |  |  |
|              | Changed to "To determine relative electronic drifts, the DLEDs which are situated close to the detectors are used. The optical path of the  |  |  |
|              | WLS includes additional elements which are not part of the optical path for light from the Earth or the Sun, and the WLS light does not     |  |  |
| p 15 l286 ff | pass through the QVDs. The internal light sources also show a decrease in output which cannot be separated from instrument                  |  |  |
|              | degradation as described in Section 4. The internal light sources are therefore less suitable for the calibration of the degradation of the |  |  |
|              | irradiance and radiance optical paths.                                                                                                      |  |  |
|              | Changed to "Radiance measurements in general show much variability in themselves and would require too much input from                      |  |  |
| n 15   287   | atmospheric models to be useful for the derivation and regular update of an independent and sufficiently accurate degradation               |  |  |
| p 131207     | correction for operational L1b processing. In the future the derived correction needs to be validated by - for example - using sites with   |  |  |
|              | well known reflectance."                                                                                                                    |  |  |
|              | Changed from "the degradation of the diffusers (QVD1 and QVD2) used for irradiance measurements, a gradual spectrally dependent             |  |  |
| p 15   291   | increase of the throughput in the UV spectrometer and a drift of the CCD gain for the UVN spectrometers." To "the degradation of the        |  |  |
|              | diffusers (QVD1 and QVD2) used for irradiance measurements, a drift of the CCD gain for the UVN spectrometers and a gradual spectrally      |  |  |
|              | dependent increase of the throughput in the UV spectrometer. This spectral ageing in the UV spectrometer is observed for irradiance,        |  |  |
|              | radiance and WLS data and cannot be found in on-ground data."                                                                               |  |  |
| p 16   300   | Changed "composed" to "modelled"                                                                                                            |  |  |
| p 16   304   | Changed "perfect" to "are best described"                                                                                                   |  |  |

| Item         | Change                                                                                                                                                   |  |
|--------------|----------------------------------------------------------------------------------------------------------------------------------------------------------|--|
|              | Added "For UVN (SWIR) a super pixel stretches over 20 (12) rows in the spatial direction. In the spectral direction (columns) it is 5,10,20              |  |
| p 16   315   | and 20 pixels for UV, UVIS, NIR and SWIR respectively. Apart from the spectrometer degradation in the UV, the data is spatially and                      |  |
|              | spectrally smooth, so the super-pixel size has no impact on the result apart from noise reduction."                                                      |  |
| p 16   316   | Added "Following the postulate of the model, the"                                                                                                        |  |
| p 16   318   | Added "If the residuals show in the future that the assumption of exponential decay is not justified anymore, a different fitting function can be used." |  |
|              | Rephrased to "In the left part of Fig.13 it can be seen that this spectrometer ageing is stronger than the signal decrease due to the                    |  |
| p 16   333ff | diffuser degradation. In this way the UV spectrometer ageing nullifies the diffuser degradation."                                                        |  |
| p 17   347   | Added "diffuser"                                                                                                                                         |  |
|              | Added "The spectrometer specific degradation Dspec in the UV spectrometer is derived for the entire mission so far and the correction is                 |  |
| p171349      | applied to both the radiance and irradiance. The correction is also applied to the reference orbits for the absolute irradiance calibration"             |  |
| p 17   350   | Added " and that the steps occurring in the data around updates are minimal."                                                                            |  |
| n 22 Table 4 | Changed "The degradation per band per 1000 orbits as determined up to orbit 9748" to "The mean degradation per 1000 orbits as                            |  |
| p zz Table 4 | determined up to orbit 9748." Added % to the header.                                                                                                     |  |
|              | Added "An investigation of various on-ground illumination sources via the Sun and the Earth port showed that the discontinuity is                        |  |
| p 23 l 362   | exclusively observed for the absolute irradiance calibration with the FEL lamp. The absolute radiance calibration with the FEL lamp is                   |  |
|              | consistent with other calibration sources."                                                                                                              |  |
| n 23   364   | Added "The correction to the absolute irradiance is derived for orbits 2818 (QVD1) and 2819 (QVD2), the same orbits the diffuser                         |  |
| p 23 1 30 4  | degradation is tied to. The UV spectrometer specific degradation has been corrected in the used data, see Section 12."                                   |  |
|              | Changed "A well-known solar reference is the high resolution Dobber spectrum (0.014 nm per pixel) (Dobber et al., 2008) and the Kurucz                   |  |
|              | spectrum (Chance and Kurucz, 2010), which are high resolution composites of different solar measurement campaigns. It covers the                         |  |
| n 23   365   | spectral range of the TROPOMI instrument, but especially in the UV range it is unclear if it is reliable." To "Well-known solar references               |  |
| p 20 1 000   | are the high resolution Dobber spectrum (±0.014nm per pixel) (Dobber et al., 2008) and the Kurucz spectrum (Chance and Kurucz, 2010),                    |  |
|              | which cover the spectral range of the TROPOMI instrument. They are both high resolution composites of different solar measurement                        |  |
|              | campaigns and not based on a single instrument."                                                                                                         |  |
| p 23   375   | Removed "independently calibrated"                                                                                                                       |  |
| p 23   376   | Added two references: Seftor et al., 2014; NASA Goddard Space Flight Center, 2019                                                                        |  |
| p 23 l 380   | Replaced "spectral" by "radiometric"                                                                                                                     |  |
|              | Added "Adapting only the irradiance calibration for UV and UVIS changes the reflectance for these spectral ranges. Initial validations                   |  |
| p 24 l 402   | tests show that this has indeed a positive impact on the L2 retrievals. In the future a more extensive re-assessment of the radiometric                  |  |
|              | accuracy can be performed and any potentially remaining inconsistencies in radiance and irradiance can be addressed."                                    |  |

| Item         | Change                                                                |  |
|--------------|-----------------------------------------------------------------------|--|
| p 24 l       | Adapted SSD to be consistent with table: 5.6-> 5.5, 7.2 ->7, 28.8->28 |  |
| 423/424      |                                                                       |  |
| p 25, l 434/ | Changed v1 / v2 to version 1 /2                                       |  |
| 436          |                                                                       |  |
| p 449        | Changed "radiometry" to "radiometric"                                 |  |
| References   | Removed urls where doi is present, removed doi prefix in bib-file.    |  |
| Language     | Removed phrase "corrected for".                                       |  |

**Changes to initial version 1**

The changes below have been performed to the initial version 1 sent out to the referees. These changes were already included in the version 2 which was published on the discussion page and are listed below for completeness.

| Line number Fig/Table
(version1/version2) | Original (version 1)                                                                                                                                                                                                                          | Update (version 2)                                                                                                                                                                                                                                                                                  |
|----------------------------------------------|-----------------------------------------------------------------------------------------------------------------------------------------------------------------------------------------------------------------------------------------------|-----------------------------------------------------------------------------------------------------------------------------------------------------------------------------------------------------------------------------------------------------------------------------------------------------|
| /Table 1                                     |                                                                                                                                                                                                                                               | Added table on main characteristics.                                                                                                                                                                                                                                                                |
| 24/24                                        |                                                                                                                                                                                                                                               | Added " The main characteristics of TROPOMI are listed in Table 1. "                                                                                                                                                                                                                                |
| 23/23                                        | 5.5km x 3.5km                                                                                                                                                                                                                                 | Put non-rounded number to be consistent with new table: 5.6km x 3.6km                                                                                                                                                                                                                               |
| 420/423                                      | "before it was approximately 7 km
at nadir and it is now about 5.5 km.
In across-track direction the minimal
sampling distance
at nadir is around 3.5km for bands
2–6, about 7km for bands 7–8 and
around 28km for band 1." | Put non-rounded number to be consistent with new table: "before it was
approximately 7.1km at nadir and it is now about 5.6km. In across-track direction the
minimal sampling distance
at nadir is around 3.6km for bands 2–6, about 7.2km for bands 7–8 and around 28.8km
for band 1." |
| Caption Fig.5/Fig.5                          | "The differences for low and high
row numbers are now mostly within
the requirements and more
symmetrical."                                                                                                                          | Added "(black lines)":
"The differences for low and high row numbers are now mostly within the
requirements (black lines) and more
symmetrical."                                                                                                                                           |
| 208                                          | 335337                                                                                                                                                                                                                                        | Changed to em-dash: "335–337"                                                                                                                                                                                                                                                                       |
| Caption Fig.6/Fig.6                          | "Note that the row numbering is showing the binned count."                                                                                                                                                                                    | Changed to "Note that the binned row count is shown in the plots, the affected detector rows are rows 335337."                                                                                                                                                                                      |

| Line number Fig/Table | Original (version 1)                   | Update (version 2)                                                                           |
|-----------------------|----------------------------------------|----------------------------------------------------------------------------------------------|
| (version1/version2)   |                                        |                                                                                              |
| 220/221               |                                        | Added: "Detector rows 335 and 336 correspond in this example to the binned row               |
|                       |                                        | counter 144."                                                                                |
| 263/265               | "For double processing, so re-         | Re-phrased to: "To validate the integration of processor and key data, double                |
|                       | analysing data that is corrected with  | processing is performed: data that has already been corrected with the derived CKD is        |
|                       | the derived relative irradiance CKD,   | re-analysed for remaining effects. Double processing irradiance data with the derived        |
|                       | the standard deviation reduces to      | relative irradiance CKD reduces the standard deviation to the order of $\times 10$ –4 . This |
|                       | the order of ×10 −4 , this is an order | result is an order of magnitude better than what was achieved with double processing         |
|                       | of magnitude better than what was      | of the CKD derived from on-ground calibration data."                                         |
|                       | achieved with the on-ground data."     |                                                                                              |

---

## Author Response (AR1)

**Review comment amt-2019-488-RC1**

**Reviewer: **Anonymous Referee #3**

Dear referee,

Thank you for your detailed review of our article. Our responses to your remarks, questions and considerations can be found in the table below. The performed changes to the manuscript are listed in the Section "Detailed Changes".

**Response**

| Item | Referee comment | Author's response |
|------|-----------------|-------------------|
| Section 5 | The text states that in-flight linearity deviates from on-ground by no more than 1%. This seems rather large. Is this a statistically significant deviation? This deserves more discussion. | The text states 1‰, which is not large. Have you maybe misread the sentence? |
| Section 7 | This is an important topic, but the authors choose to devote only a short qualitative discussion to it. It would be helpful to the reader to provide some idea of the errors involved. At what error level does the flagging occur? | Added sentence on level of saturation flagging. As described, the blooming flag is not based on an error threshold but on a pixel filling value. |
| Section 8 | The authors state they have only addressed geolocation in Bands 4-7. Geolocation in the shorter bands, esp. Band 3, are also important and validation should be possible except for Band 1. The authors should at least discuss what their plans are to validate these bands. | A more detailed discussion on the results, their consequences and future plans has been added. |
| Section 9 | A similar comment about wavelength registration. The authors imply there is no source of wavelength information other than from L2 products and there are no products providing this information for Bands 1 & 2. Yet the spectral registration in these bands is no less important than at longer wavelengths. The authors can at least acknowledge the problem and discuss their plans to deal with it. | The L1 wavelength assignment is based on on-ground calibration. The key data can be updated as described. Added specifically that this can also be done once data for other bands is available. |
| Section 11 | The discussion in this section (esp. the paragraph starting at line 260) was somewhat confusing. The authors should consider two alternatives | This section has already been adapted following the initial review comments. |

| Item | Referee comment | Author's response |
|---|---|---|
| | to remedy this: provide a bit more explanation to the reader, or eliminate some of the details that are the source of the confusion. I recommend the latter because it's not clear what is to be learned from these details. | |
| Section 12 | In Line 285 the authors seem to throw cold water on any technique, other than on-board calibrations, to derive or validate radiometric change. It is quite reasonable that the authors have not had a chance to implement any of the well-documented techniques for validating the calibration, but they should refrain from suggesting these were omitted because they lack useful information. I think I follow the 'competing change' argument described in Lines 330-335, but I doubt most readers will. The authors need to describe explicitly what about Figures 12 & 13 indicates increasing detector response competing with diffuser degradation. | Rephrased the sentence to make clear that it is about the operational L1b processor. Added a remark on validation of the correction with Earth targets.

Rephrased to make the point clearer. |
| Table 3 | These numbers appear to be in percent. The authors should say so explicitly | Added % to the header. |
| Section 13 | The authors imply at the start of Section 12 that the reflectance calibration of TropOMI is an important quantity, but they fail to address its accuracy. If that is outside the purview of this paper, the authors should say so. The authors also fail to discuss in this section the effect that adjusting the irradiance calibration has on measured Earth TOA reflectances. Since the radiance calibration wasn't mentioned, the reader is left to assume that all the adjustments described in Section 13 are being applied in inverse to the instrument's reflectance calibration. What is the justification for doing so? The authors provide no insight as to why the pre-launch irradiance calibration might be so much in error. How do they know that the radiance calibrations are not in error by an equal or nearly equal amount? | Added explicitly that the reflectance is changing. As described in the beginning of Section 13, the on-ground calibration measurements for irradiance suffered from low SNR. As mentioned, the details about the on-ground calibration issues are discussed in Kleipool et al. (2018). Added more details on further comparison with on-ground sources. |
| Grammar comment | Use of the word "for" in connection with "corrected" should be accompanied by an object rather than a subject. "We correct for something" rather than "Something is corrected for." | Adapted. |

**Review comment amt-2019-488-RC2**

**Reviewer: **Anonymous Referee #2**

Dear referee,

Thank you for your detailed review of our article. Our responses to your remarks, questions and considerations can be found in the table below. The performed changes to the manuscript are listed in the Section "Detailed Changes".

**Response**

| Item | Referee comment | Author's response |
|---|---|---|
| Fig 1-3,5,6,11-22 | The used fonts are to small, enlarge | The plots have been adapted or enlarged. Captions have been adapted accordingly. |
| Fig. 7-9 | At least the numbers at the color scales need to be larger. For not so young eyes, the numbers and labels in the printed paper are difficult to read. Please change. | The plots have been adapted or enlarged. Captions have been adapted accordingly. |
| p1 l13, abstract | "processing from 2020 on". From my knowledge, late 2020 is the current foreseen start for the version 2 L1b processor. Please adapt the date, also in the conclusions. | Yes, that is correct. Currently the planning is for *late* 2020. Adapted both occurrences. |
| p2 l41, introduction | please define the term "orbit types" (probably the measurement sequence along the orbit.) | Adapted. |
| P3 , introduction | Please add (at the end of the introduction) a sketch of the instrument design, it would be very useful for the following paragraphs: Where is the calibration unit, what are the light paths, where are the diffuser etc. Please also add a paragraph or a sketch to the detector layout: row and column is frequently used in the text, but nowhere the spatial and spectral direction is explicitly stated. | Added a new figure with the functional schematic of TROPOMI. The spatial and spectral direction is now added to the text several times. |
| Section 2 Thermal stability | It is observed, that the thermal stability is reduced after orbital manoeuvres. Is there a reason or at least an educated guess for this behaviour? If yes, please add. | Added. The thermal stability is reduced when the pointing of the radiant cooler is not optimal as can be the case during manoeuvres. |

| Item | Referee comment | Author's response |
|------|-----------------|-------------------|
| Section 7 Pixel saturation, blooming | Are there estimations available, how frequently saturation and charge blooming occur? Which are the suspect conditions (snow? tropical clouds? something else?). Please add. | Information on the pixel saturation has been added. For the blooming we do not have detailed statistics yet as the new version is not in use yet. |
| Section 8 Geolocation | p9 l161: ..or along-rack.. -> or along-track | Adapted. |
| Section 9 Spectral annotation | It is stated, that the calibration key data for the wavelength calibration are updated according to the wavelength fits in the Level 2 algorithms. Are the key data directly used as wavelength axis? There is no Level 1 wavelength calibration? | Currently there is no online Level 1 wavelength calibration. The L1 key data is based on on-ground calibration and adapted with the in-flight insights. L2 retrieval algorithms perform their own wavelength fitting where the L1 wavelength assignment is used as a starting point. |
| Section 10 Slit irregularity | Especially for this section, the definition of rows/columns versus spatial/spectral direction in the introduction would be very useful! | Adapted. |
| Figure 6 | Change y-axis name to 'binned row counter' (this is the used name in the text). Please add in the caption, that the row 335-337 corresponds to the binned counter 144. | This has already been adapted following the initial review comments. |
| p 16 l 303 ff Section 12 absolute radiometry and instrument degradation | "The specific degradation curves ... are perfect exponential curves". Here it is assumed, that the degradation behaves exponential, so write something like: "It is assumed, that the diffuser degradation behaves exponential with time. Therefore, the specific degradation curves ... are modelled as exponential curves". Also the exponential behaviour of $D_{com}$ is an assumption and should be stated as such. | It was found that the exponential fits resulted in a better fit than other functions. We made this clearer in the text that this is the model. |
| P 16 l 315 ff Section 12 absolute radiometry and instrument degradation | "For each of these super pixels the linear system in Eq. (1) is solved. For the UVIS, NIR and SWIR no spectrometer degradation $D_{spec}$ could be determined and this term is therefore set to unity." I think, this is the wrong order: For UVIS, NIR, and SWIR, no spectrometer degradation can be derived, therefore $D_{spec}$ is set to 1.0 for theses channels. With this assumption, the linear equations system is solved for each super pixel if UVIS, NIR and SWIR. Right? Please also give a number, how many pixels are in one super-pixel. | For UVIS, NIR and SWIR no spectrometer degradation was found so the term was set to 1.

Added the size of the super pixels and some more explanation on it. |

| Item | Referee comment | Author's response |
|---|---|---|
| p 16 l 317 Section 12 absolute radiometry and instrument degradation | "The solutions for $D_{q1}$, $D_{q2}$ and $D_{com}$ are all three exponential decay functions and perfectly smooth in the temporal dimension." Your model fits exponential decay functions for this quantities, therefore this is trivial message. What could be stated here is something like: The assumption of an exponential decay for $D_{q1}$, $D_{q2}$ and $D_{com}$ is approved by the small residuals $R_k/P_k$, as shown by the right plots in Fig 12-14. The explanation for estimating $D_{spec}$ for the UV leaves a few questions open: $D_{com}$ is extrapolated to the UV region. What about $D_{q1}/D_{q2}$? What type of extrapolation do you use, so what are the assumptions made? Towards shorter wavelengths, the degradation is expected to increase. According to the left plot in Fig. 11, this is not the case for $D_{spec}$. | The phrasing has been changed to make clear that the exponential decay is the model.

For the ratio of Dq1/q2 the assumption is made, that the degradation is exposure based, so we use the total time of usage as input. The spectral ageing Dspec of the UV spectrometer does indeed behave different than the diffuser degradation. The signal increases with time. |
| P 17, l 347-352 | For the forward processing, an extrapolation of the degradation parameters is used. It is stated, that this new degradation parameters will be regularly updated by incorporating the recent measurements. With the update, also the extrapolation will change. This might introduce jumps in the irradiance time series, which might be an issue for users. Is there a strategy to monitor and/or avoid this? Please add some information about the details here. | Added remark on jumps in the data. |
| P 22, Table 4 | The *mean* degradation per Band is given, right? Please clarify. | Yes for the bands/ wavelength it's the mean degradation. Rephrased the caption. |
| 13 Absolute irradiance calibration: | Why is the OMPS irradiance measurement choosen as the reference measurement for the radiometric calibration? To my knowledge, OMPS is an unusual solar reference measurement. OMPS does not even distribute there irradiance measurements as regular product. The cited literature [Jaross 2014] gives no information about the absolute radiometric calibration except a plot together with an unnamed "synthetic" spectrum. If possible, at a reference for the radiometric calibration of the OMPS irradiance. Recently re-calibrated and published solar spectra are SOLSPEC (Meftah et al 2018) or SCIAMACHY (Hilbig et al, 2018), which would be a better choice. Both are also independent from other reference spectra. Nothing is said about the radiance calibration. Is the discontinuity observed in the overlap region also visible in radiances? What about the reflectance? The light path is the same for radiance and irradiance except the QVD. The QVD is the same for the the UV / UVIS overlap and cannot cause the discontinuity. Therefore, in the reflectance the discontinuity should | The OMPS solar irradiance is not distributed as a separate product, but is part of every L1b file, we added additional references.
We now added explicitly that the reflectance is changing, and explained the observed inconsistencies from on-ground calibration.
The OMPS spectrum was chosen for several reasons: it has similar instrument characteristics, it is an active mission and a single instrument spectrum and not a composite spectrum. This point is made clearer in the text now.
As shown in the paper we compared the results to different references. |

| Item | Referee comment | Author's response |
|------|-----------------|-------------------|
| | cancel. If only the irradiance is mitigated here, the discontinuity is introduced in the reflectance. The radiance calibration and the impact of the irradiance mitigation on the reflectance needs to be discussed here. | |
| Conclusions p 25, l 434/ 436: p 25, l 449 | 'v1' / 'v2' change to 'version 1' / 'version 2', radiometry -> radiometric | Adapted. |
| References | Many references contain both the the DOI based URL and a direct URL. Only the DOI URL as permanent URL is needed, skip the second URL (which is also not added consistently...). | Corrected. |
| References | For Ingmann et al. The URL is erroneous | Corrected. |

**Review comment amt-2019-488-RC3**

**Reviewer: **Rüdiger Lang**

Dear referee,

Thank you for your detailed review of our article. Our responses to your remarks, questions and considerations can be found in the table below. The performed changes to the manuscript are listed in the Section "Detailed Changes".

**Response**

| Item | Referee comment | Author's response |
|------|-----------------|-------------------|
| Section 12/13 | Section 12 and 13 describe the approach taken to correct for some partially significant, observed degradation effects especially in the UV. The overall approach seems sound (section 12). However it is not obvious for me how the degradation model approach and application in section 12 is related, or better decoupled, from the correction of the observed, partially quite significant offsets (up to 15%) in the absolute irradiance calibration of the solar port (section 13). My | From the on-ground calibration we are sure that the irradiance calibration is not correct for bands 1-3 (made this clearer in the text). For the on-ground calibration of the radiance there is no such evidence. By setting the reference for the (spectrally smooth) diffuser degradation and the absolute irradiance |

| Item | Referee comment | Author's response |
|------|-----------------|-------------------|
| | understanding from the paper is that the derived spectrometer component (from the 312 to 330 nm region) has been accounted for by a degradation correction, which is, again to my understanding, applied spectrally neutral to the full UV detector irradiance. Is this correction then also applied for Earthshine measurements, as one would expect it to be, because it is considered an effect of the common optical path? In case yes, I guess that the normalization day/orbit 2818/2819 is then used for an adjustment to OMPS, such that any likely degradation happening to the irradiance signals until this point is corrected for by reference to OMPS. Again, one expects an unknown degradation to have happened also to the Earthshine path until orbit 2818/2819, which would then lead to a differential degradation in reflectance after adjustment of the solar irradiance, and especially in case nothing is done additionally for the Earthshine data (and probably there are also some finite yet different accuracies for the radiometric key-data to be taken into account). | adaptions on the same orbit, the diffuser degradation up to that point is taken into account. The spectral ageing in the UV spectrometer is corrected in radiance and irradiance for the entire mission. The spectral features already present in orbit 2818 are therefore removed and the smooth correction of the absolute irradiance takes care of the diffuser degradation. We made this point clearer in the text.
If there is a remaining inconsistency in radiance this needs to be addressed in future validation for example via Earth targets, this has also been added to the text. |
| Section 12/13 | The choice of OMPS seems also very subjective. While it is stated that OMPS irradiance has been "independently calibrated", it is not stated what "independently" would mean in this context (without adjustment to reference spectra? If yes, then this should be stated). I would maintain that it remains just a choice. The results show a close to 3% difference with the Dobber et al. spectrum after adjustment. In contrast, all three GOME-2 instruments shave shown smaller residuals than 3% to the Dobber reference spectrum, above 300 nm at the beginning of live, without (!) adjustment (so using the on-ground derived key-data only). So this choice of a reference solar spectrum would leave a potential unknown "offset" of 2 to 3% with respect to other instruments and their absolute calibration after degradation correction. Since 2 to 3% accuracy is effectively the current limit on the knowledge of the solar irradiance accuracy in the UV and VIS wavelength region in general, such a choice for sure can be made, but it should be presented as the limit of the knowledge in the absolute calibration accuracy then also for this mission. Moreover, this would then also be the limit of knowledge on the Earthshine radiance accuracy, with a potentially even larger error on the reflectance. In this respect, the question is why an independent Earthshine degradation modelling has been ruled out. For previous missions GOME-1, 2 and | We made clearer in the text why OMPS was chosen. The idea was to be able to relate the changes to a single instrument and not a composite spectrum. And indeed, eventually it is a choice.
We added a clarification on validation using Earth targets, this is future work. Therefore we are also not presenting any updated numbers for the radiance and reflectance accuracy yet. |

| Item | Referee comment | Author's response |
|---|---|---|
| | SCIAMACHY degradation modelling using global averages of cloud free Earthshine data showed quite some success, and also Libyan desert degradation modelling should not be ruled out. | |
| Section 12 | Finally, the derived spectrometer component in Section 12 seems to be on the order of 1% per 1000 orbits (Figure 11). In contrast the observed WLS and LED signal degradations seem to be lower or on the same order. I am wondering why the use of the internal light sources then have been ruled out for degradation monitoring or even correction, or how their "output degradation" could have been identified as such, when the identified spectrometer component is on the same order or even more significant. Is there an optical component in the path (like another folding mirror) between the spectrometer and the WLS, such that any direct Earthshine degradation modelling using these sources cannot easily be done? It might be interesting to look at the ratio of calibrated SMR and calibrated WLS, and their (differential) evolution over time and spectrally in this context. | Analysing WLS data has given valuable insight on the spectral ageing in the UV and the spectral overlap with UVIS. The light path of the WLS includes additional optics and is not identical to the Sun or Earth path. When using WLS for calibration purposes WLS features could be introduced into the L1 radiance/irradiance.

Added an explanation to the difference in light paths for DLED and WLS |
| Section 3, l 80ff | How exactly non-linear is the observed decrease of the light sources and can this decrease be attributed to the sources or is it already part of the optical chain for WLS? It should not be ruled out that this is simply a consequence of the spectrometer degradation observed in Section 12 (see before). | The decrease in signal can be caused by the source itself, the source's specific optical path and the instrument. Clarified this in the text. |
| Section 5, l.90ff | I would assume the temperature dependency of the dark current has been measured on-ground. From these measurements it could be stated here what is the projected dark current orbital dependency using the observed orbital detector thermal stability from HKTM. | The temperature dependency of the dark current has not been measured on-ground. |
| L.110ff: | The change in the gain during manoeuvres is not further explained. Can any reason be given for this? | So far no reason has been found to explain this behaviour. All available housekeeping parameters have been checked but no correlation was found. |
| l. 145ff | It would be interesting (and helpful for future missions) to get an idea (statistically) on the extend of blooming in pixel space. E.g. by providing a histogram (or table) on the number of occurrences over the number of pixels affected per event. Does such a statistic exist? | We have added numbers for the occurrence of pixel saturation. For the blooming itself a full statistical analysis is possible once version 2 of the L01b processor is active. |
| Section 8 on geo-referencing | Has any attempt be made for geo-rectification using VIIRS data? This should provide very accurate geo-referencing knowledge also on the point-spread function. Can anything be said about the alignment of the other bands not used in | We have not attempted any cross-validation of the geolocation with VIIRS or other satellites. Considering the limited spatial resolution of TROPOMI we don't |

| Item | Referee comment | Author's response |
|---|---|---|
| | the geo-referencing analysis? Or can some qualitative assumption be derived from the optical setup (telescope) and alignment? A discussion would be needed here I think. | think that a comparison to higher resolution instruments would have added to the results. A discussion on qualitative assumptions for the other bands has been added. |
| Section 10 on slit irregularities | From Figure 6 it looks like the WLS exhibits significant spectral structure. Why is this? Actually, wouldn't a highly structured spectrum like the solar lead to a better correction? | Figure 6 is an irradiance image showing characteristic spectral lines. The data has been corrected with key data derived from WLS data. |
| Section 11 on goniometry | The azimuthal maximum variation of the sun should be reported in this Section in order to motivate/justify the restriction to 10 degrees, even though 15 degrees have been measured. Is the orbit stabilized, and for how long in the mission? Or in other words, is there any restriction in future ground track drifts concerning the validity range of this data? | The natural solar azimuth range during solar calibration measurements over one year is between -10° and +6.0° (range is 16°) for an ANX MLST of 13:30, which is the current mission requirement. The instrument requirements were therefore set to allow for measurements in the range -10° to +10° in azimuth and -4.25° to +4.25° in elevation. In reality the solar baffle allows light for a larger range, the measurements were therefore performed at the largest possible range achievable with the platform. The orbit is stabilized and follows Suomi NPP with a 3 to 5min delay. For nominal operations the solar port should not run out of its calibrated range. Furthermore the irradiance measurements are performed around a fixed azimuth angle using the reaction wheels. This is explained in Section 14, we added a reference to this and a sentence on the natural azimuth range. |
| Section 11 | Section 11, on the origin of the remaining residuals in the goniometry key-data derived in-flight: I would guess that they are probably a combination of diffuser features, speckles, and especially instrument drifts between individual measurements and the temporal position of the normalization measurement. In addition, one should find the pattern of the observed degradation correction residual in such a potential drift, I would assume. Since the measurement period was quite long (400 orbits), and it was in an early state of the mission, can effects | The remaining residuals are - as you observe - connected to the residuals observed in the degradation correction. Added a clearer reference to the residual discussion in Section 12. Thermal effects can be excluded, the instrument was thermally stabilized since very early in the commissioning phase. The main part of electronic gain drifts is |

| Item | Referee comment | Author's response |
|---|---|---|
| | like gain drifts during this period, and as reported in the earlier sections, be ruled out? It would be good to discuss the status of the mission at the time of the dedicated measurement period (start orbit, overall platform thermal stability etc...), and if the measurements have been filtered for outliers. | corrected by the use of the normalization measurements.
Added start orbit, remark on electronic drifts and in Section 2 a remark on thermal stability for the measurements. |
| Section 12, degradation model: | Why would one expect that all components are "perfectly exponential". At least in the long-run. Since this is not what is observed with other instruments, and for sure not in case of a potential mirror contribution. Is there a long-term trend observed in the Rk and Pk components? | Until now an exponential function was found to be the best fit. If this changes in the future we can adapt it.
Added this remark to the text. |
| l. 364 | "but especially in the UV range it is unclear if it is reliable": Which spectrum is referred here to? Since we have observed that the Dobber et al., spectrum shows clearly better results for GOME-2 for wavelength below 300 nm at the beginning of the mission and without any adjustments than all other available reference spectra. I fear that at this stage this is no discussion about the truth, but probably more about inter-instrument consistencies. | That is the difficulty with inter-instrument comparisons, in the end it is a matter of choice. We tried to base our choice on how comparable the spectral resolutions and ranges are and that the we are traceable to a single instrument and not a composite. This point was made clearer in the text. |
| General: | Although it has been describe multiple times elsewhere, a table of band numbering associated with source region "UV", "UVIS", "NIR" "SWIR" and associated wavelength ranges would be of help for the reader to have at hand up-front. Since band numbers, detector labels and source regions are used multiple times in exchangeable ways in the paper. | This has already been adapted following the initial review comments. |
| Figure 5/6 | "...within the requirements" -> add black lines in brackets p. 10ff: The plots in Figure 6 and the reported row numbers in the text (e.g. line 208) are different. The caption indicates the Figure shows the binned count. Somewhere at least a written translation should be made. E.g. in the caption: bin x corresponds to pixels yy. Or similar. | This has already been adapted following the initial review comments. |
| p14, ;l263: | Check sentence: "For double processing, so (?) ..." l.380 switch -> with | This has already been adapted following the initial review comments. |

**Detailed changes**

**List of changes to version 2**

The page and line numbering in the Table below is according to version 2 which was public on the discussion page. The comments on the version 1 (the one which was initially sent out to the reviewers) have already been included in version 2.

| Item | Change |
|---|---|
| New figure | Added new figure and caption at the beginning of the article. It shows a functional schematic of TROPOMI. Added a reference to this figure in several places in the text. |
| Fig 1-3, 5-9, 11-22 | Enlarged plots or adapted plots to increase fontsize and improve readability. Adapted captions and the references to the plots accordingly. For Fig.1 added "The triangles in the top panel show the gain ratio as derived from on-ground measurements." |
| p 1 l 13, abstract | Changed "processing from 2020 on" to "processing from late 2020 on". |
| p 1 l 22/23, Table 1 | Adapted to be consistent with official PRF: 5.6-> 5.5, 7.2 ->7, 28.8->28 |
| p2 l 27 ff | Replaced "The instrument is measuring the radiance on the day side of each orbit and once a day the irradiance via a dedicated solar port as described in detail in KNMI (2017) and Kleipool et al. (2018)." By " The instrument is measuring the radiance on the day side of each orbit and once a day the irradiance via a dedicated solar port as shown in Fig.1. Sun light passes through one of the two internal quasi volume diffusers (QVD1 and QVD2) and is coupled via the folding mirror into the telescope of the instrument. A detailed instrument description canbe found in KNMI (2017) and Kleipool et al. (2018)." |
| p 2 l 41, introduction | Changed "The timing and definition of the different orbit types was adapted to match the detected darkness of the eclipse. " to "The timing and definition for the measurement sequences of the different orbit types was adapted to match the detected darkness of the eclipse. " |
| p 3 l58 | Added " All measurements described in this article were performed at the nominal temperatures with active thermal stabilization." |
| p 3 l 62 | Added "when the radiant cooler points in a sub-optimal direction" |
| p 4 l 75 ff | Replaced "output" by "observed signal"and "detector response". |
| p 4 l 86 | Added "Depending on the source and its location in the instrument, the listed values can contain contributions from degradation of the source, its specific optics, the diffusers, the folding mirror, the telescope and the spectrometers." |
| p 6 l 114 | Added "or other housekeeping parameters" . |
| p 7 l 138, 141 | Added " in the tropics". Changed "and" to "-". Added " In the tropics typically about 0.2-0.5% of the pixels are flagged for saturation in bands 4-6, other regions and bands are hardly ever affected." |
| p7 l 140 | Added: " For the CCD detectors spatial binning is applied: the charge of several successive detector rows is added in the register and then read out." |

[revised manuscript text omitted]

| Item | Change |
|---|---|
|  | which cover the spectral range of the TROPOMI instrument. They are both high resolution composites of different solar measurement campaigns and not based on a single instrument." |
| p 23 l 375 | Removed "independently calibrated" |
| p 23 l 376 | Added two references: Seftor et al., 2014; NASA Goddard Space Flight Center, 2019 |
| p 23 l 380 | Replaced "spectral"by "radiometric" |
| p 24 l 402 | Added "Adapting only the irradiance calibration for UV and UVIS changes the reflectance for these spectral ranges. Initial validations tests show that this has indeed a positive impact on the L2 retrievals. In the future a more extensive re-assessment of the radiometric accuracy can be performed and any potentially remaining inconsistencies in radiance and irradiance can be addressed." |
| p 24 l 423/424 | Adapted SSD to be consistent with table: 5.6-> 5.5, 7.2 ->7, 28.8->28 |
| p 25, l 434/ 436 | Changed v1 / v2 to version 1 /2 |
| p 449 | Changed "radiometry" to "radiometric" |
| References | Removed urls where doi is present, removed doi prefix in bib-file. |
| Language | Removed phrase "corrected for". |

**Changes to initial version 1**

The changes below have been performed to the initial version 1 sent out to the referees. These changes were already included in the version 2 which was published on the discussion page and are listed below for completeness.

| Line number Fig/Table (version1/version2) | Original (version 1) | Update (version 2) |
|---|---|---|
| /Table 1 |  | Added table on main characteristics. |
| 24/ 24 |  | Added " The main characteristics of TROPOMI are listed in Table 1. " |
| 23/23 | 5.5km x 3.5km | Put non-rounded number to be consistent with new table: 5.6km x 3.6km |
| 420/423 | "before it was approximately 7 km at nadir and it is now about 5.5 km. In across-track direction the minimal sampling distance | Put non-rounded number to be consistent with new table: "before it was approximately 7.1km at nadir and it is now about 5.6km. In across-track direction the minimal sampling distance at nadir is around 3.6km for bands 2–6, about 7.2km for bands 7–8 and around 28.8km for band 1." |

| Line number Fig/Table (version1/version2) | Original (version 1) | Update (version 2) |
|---|---|---|
|  | at nadir is around 3.5km for bands 2–6, about 7km for bands 7–8 and around 28km for band 1." |  |
| Caption Fig.5/Fig.5 | "The differences for low and high row numbers are now mostly within the requirements and more symmetrical." | Added "(black lines)": "The differences for low and high row numbers are now mostly within the requirements (black lines) and more symmetrical." |
| 208 | 335-–337 | Changed to em-dash: "335–337" |
| Caption Fig.6/Fig.6 | "Note that the row numbering is showing the binned count." | Changed to " Note that the binned row count is shown in the plots, the affected detector rows are rows 335--337." |
| 220/221 |  | Added: "Detector rows 335 and 336 correspond in this example to the binned row counter 144." |
| 263/265 | "For double processing, so re-analysing data that is corrected with the derived relative irradiance CKD, the standard deviation reduces to the order of ×10 −4 , this 
[revised manuscript text omitted]